# Nodal immune flare mimics nodal disease progression following neoadjuvant immune checkpoint inhibitors in non-small cell lung cancer

Tina Cascone [1 ✉], Annikka Weissferdt[2,3], Myrna C. B. Godoy[4], William N. William Jr.[1,5], Cheuk H. Leung[6], Heather Y. Lin[6], Sreyashi Basu [7], Shalini S. Yadav[7], Apar Pataer[3], Kyle G. Mitchell[3], Md Abdul Wadud Khan [8], Yushu Shi[6,9], Cara Haymaker [10], Luisa M. Solis [10], Edwin R. Parra [10], Humam Kadara[10], Ignacio I. Wistuba[1,10], Padmanee Sharma[7,11], James P. Allison[7,12], Nadim J. Ajami [13], Jennifer A. Wargo [8], Robert R. Jenq[13,14], Don L. Gibbons [1], J. Jack Lee [6], Stephen G. Swisher [3], Ara A. Vaporciyan[3], John V. Heymach [1,15 ✉] & Boris Sepesi[3,15]

Radiographic imaging is the standard approach for evaluating the disease involvement of lymph nodes in patients with operable NSCLC although the impact of neoadjuvant immune checkpoint inhibitors (ICIs) on lymph nodes has not yet been characterized. Herein, we present an ad hoc analysis of the NEOSTAR trial (NCT03158129) where we observed a phenomenon we refer to as "nodal immune flare" (NIF) in which patients treated with neoadjuvant ICIs demonstrate radiologically abnormal nodes post-therapy that upon pathological evaluation are devoid of cancer and demonstrate de novo non-caseating granulomas. Abnormal lymph nodes are analyzed by computed tomography and $^{18}$F-fluorodeoxyglucose positron emission tomography/computer tomography to evaluate the size and the maximum standard uptake value post- and pre-therapy in NEOSTAR and an independent neoadjuvant chemotherapy cohort. NIF occurs in 16% (7/44) of patients treated with ICIs but in 0% (0/28) of patients after neoadjuvant chemotherapy. NIF is associated with an inflamed nodal immune microenvironment and with fecal abundance of genera belonging to the family Coriobacteriaceae of phylum Actinobacteria, but not with tumor responses or treatment-related toxicity. Our findings suggest that this apparent radiological cancer progression in lymph nodes may occur due to an inflammatory response after neoadjuvant immunotherapy, and such cases should be evaluated by pathological examination to distinguish NIF from true nodal progression and to ensure appropriate clinical treatment planning.

---

A full list of author affiliations appears at the end of the paper.

mmune checkpoint inhibitors (ICIs) alone or in combination with chemotherapy have become the standard of care for a subset of patients with unresectable/metastatic non-small cell lung cancer (NSCLC)[1]. These agents are being tested in the neoadjuvant setting for patients with resectable NSCLC[2] with the rationale to achieve downstaging or major pathologic response (MPR, ≤10% viable tumor in the resected tumor specimens)[3] more often than chemotherapy, and potentially induce antitumor immunity. As the use of immunotherapy has become more commonplace, it has become increasingly clear that immune-based treatments may result in unconventional response patterns that are distinct from those produced by traditional cancer therapies. One potential concern with ICIs is tumor pseudo-progression, the appearance of tumor growth without true progressive disease (PD) thought to be due to increased intratumoral immune cell infiltration, which has been reported in patients with NSCLC at a rate ranging between 0.6% and 5.8%[4–6]. However, the response of mediastinal and other systemic lymph nodes to neoadjuvant ICIs is not well characterized. Distinguishing true disease progression due to the involvement of cancer in lymph nodes from false progression is critical for clinical decision making, because failure to recognize false progression could lead to delays or avoidance of potentially curative surgical resection of the primary tumor. Moreover, misinterpretation of a false positive image could lead to enhanced toxicities resulting from an inappropriately large radiation field.

A distinct phenomenon presented in cases of loco-regionally advanced and metastatic cancers following ICI therapy is the increased $^{18}$F-fluorodeoxyglucose ($^{18}$F-FDG) uptake by lymph nodes and other organs on imaging that were not involved by cancer at the time of diagnosis and deemed cancer-free upon pathological examination[7,8]. Based on our clinical observation of these occurrences in some patients with operable NSCLC treated with neoadjuvant immunotherapy on the NEOSTAR study[9], we performed an ad hoc evaluation of the radiological and pathological characteristics of this phenomenon in patients who were treated with neoadjuvant ICIs on the randomized NEOSTAR trial (NCT03158129)[9] and in a historical cohort of patients treated with neoadjuvant platinum-based chemotherapy[10] followed by surgical resection.

In this work, we report the incidence and features of the "nodal immune flare" (NIF) phenomenon, which is characterized by radiologically abnormal nodes on restaging imaging after neoadjuvant ICIs that are cancer-free and contain de novo non-caseating granulomas upon pathological evaluation. NIF appears to be related to neoadjuvant ICI therapy rather than chemotherapy and is associated with an inflamed nodal micro-environment and unique fecal microbiome genera, but not with pathological or radiological tumor responses or toxicity to immunotherapy. Because distinguishing NIF from true nodal cancer progression cannot be achieved radiologically, we suggest that suspected cases of lymph node cancer progression after neoadjuvant ICI therapy undergo invasive pathological evaluation to avoid erroneous changes in the therapeutic approach based purely on imaging.

## Results

**Apparent radiological nodal disease progression after neoadjuvant ICIs in the absence of cancer and with non-caseating granulomas.** To determine whether the administration of neoadjuvant immunotherapy is associated with unconventional radiological patterns of nodal involvement, we examined the abnormal nodes on scans in patients after they were treated with neoadjuvant ICI therapy on the randomized NEOSTAR study[9] (Table 1). We noted that among 44 patients, several patients

**Table 1 Clinical and histopathological patient characteristics in ICON and NEOSTAR cohorts.**

| Patient cohorts | ICON (chemotherapy) (n = 28) | NEOSTAR (ICIs) (n = 44) |
|---|---|---|
| | Mean (SD) | Mean (SD) |
| Age (yr) | 64.4 (8.8) | 65.6 (8.3) |
| | n (%) | n (%) |
| Sex | | |
| Female | 14 (50) | 16 (36) |
| Male | 14 (50) | 28 (64) |
| Histology | | |
| Adenocarcinoma | 19 (68) | 26 (59) |
| Squamous cell | 9 (32) | 17 (39) |
| Adenosquamous | 0 | 1 (2) |
| Clinical Stage | | |
| Stage I | 1 (3) | 23 (52) |
| Stage II | 12 (43) | 12 (27) |
| Stage III | 15 (54) | 9 (20) |
| Smoking | | |
| Never | 5 (18) | 8 (18) |
| Former | 23 (82) | 26 (59) |
| Current | 0 | 10 (23) |

n, Number of patients. ICON ImmunogenomiC prOfiling in Non-small cell lung cancer, ICIs immune checkpoint inhibitors, SD standard deviation.

exhibited abnormal nodes on imaging after treatment that mimicked disease progression but were devoid of cancer upon pathological examination and were instead characterized by the presence of non-caseating granulomas, as shown Fig. 1a–f. We referred to this phenomenon as nodal immune flare (NIF). For comparison, an illustrative case of abnormal nodes on imaging after neoadjuvant ICI therapy that was classified as true nodal disease progression based on the presence of malignant cells upon pathological assessment and the lack of non-caseating granulomas is shown in Fig. 1g–k. We then questioned whether this pattern of apparent radiological nodal progression in the absence of cancer and histological presence of non-caseating granulomas also occurred in patients that were treated with neoadjuvant platinum-based chemotherapy, although empirically we have not been alerted to this phenomenon previously. We analyzed abnormal nodes on imaging after neoadjuvant chemotherapy in a subset of patients from the ImmunogenomiC prOfiling in NSCLC (ICON) cohort at our institution[10] (Table 1) and did not observe instances where abnormal nodes post-therapy were devoid of cancer and contained non-caseating granulomas upon pathological examination. These findings suggest that neoadjuvant ICI therapy, compared with platinum-based chemotherapy, is more likely to be associated with unusual radiological appearances of cancer-free lymph nodes that mimic disease progression with the onset of pathological features of sarcoid-like inflammation.

**De novo non-caseating granulomas are a pathological hallmark feature of NIF after neoadjuvant immunotherapies.** Next, we further evaluated the cytological/histopathological characteristics of nodes collected following neoadjuvant ICIs in NEOSTAR patients and after neoadjuvant chemotherapy in ICON patients. We compared the pathological characteristics of these specimens to that of nodal samples collected at baseline invasive mediastinal staging. We noted that seven patients with abnormal nodes after neoadjuvant immunotherapy that were negative for cancer on pathological evaluation post-ICI therapy contained de novo non-caseating granulomas compared to baseline nodal specimens (Fig. 2a, b). In contrast, none of the patients treated with neoadjuvant platinum-based chemotherapy demonstrated radiologically

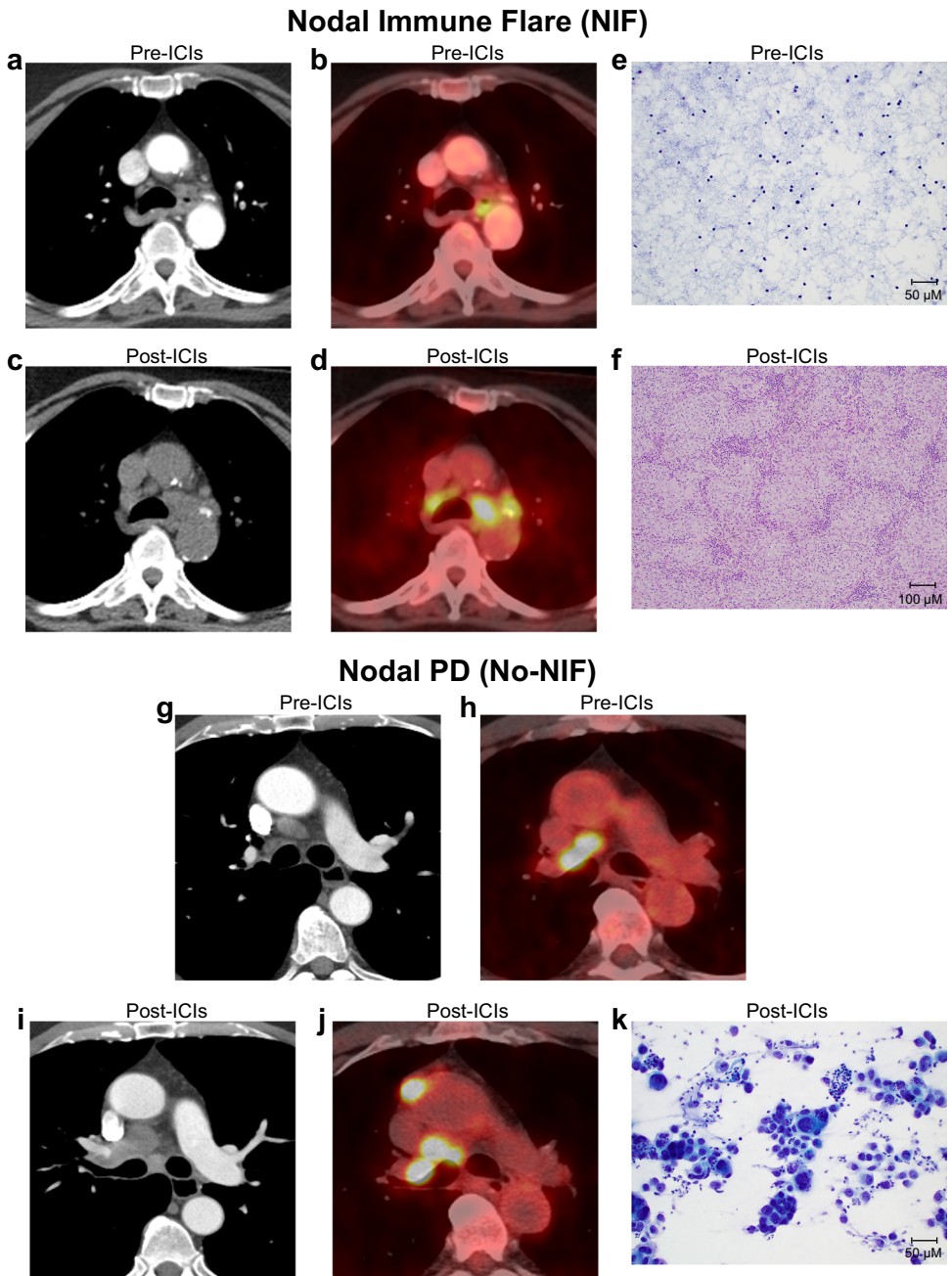

**Fig. 1 Radiological and histopathological features of abnormal nodes following neoadjuvant ICIs. a–d** Axial contrast enhanced CT (**a**), and [18]F-FDG PET/CT (**b**) images of the mediastinum showing normal nodes prior to neoadjuvant treatment with ICIs on NEOSTAR study in a patient with NSCLC (metastasis to station 7; stations 4 R, 4 L, and 11 L negative after invasive baseline mediastinal staging with EBUS). [18]F-FDG uptake in the mediastinum is due to esophagitis. Restaging axial CT (**c**) and [18]F-FDG PET/CT (**d**) images post-neoadjuvant ICIs show marked increase in nodal size and FDG uptake at bilateral mediastinal regions, suspicious for nodal progression. Mediastinoscopy post-neoadjuvant ICIs did not demonstrate carcinoma in lower paratracheal stations (4 L and 4 R). **e, f** FNA image of paratracheal nodal station pre-therapy (**e**) demonstrating lack of tumor cells and normal composition (Papanicolaou, x20), and resected station 4 R lymph node post-therapy (**f**) revealing absence of cancer and evidence of necrotizing non-caseating granulomatous inflammation (hematoxylin and eosin, x10). **g–j** Axial contrast enhanced CT (**g**), and [18]F-FDG PET/CT (**h**) images of the mediastinum show nodal enlargement and abnormal [18]F-FDG uptake in the right hilum and right mediastinum prior to neoadjuvant ICIs on NEOSTAR study in a patient with NSCLC (baseline invasive mediastinal staging with mediastinoscopy revealed metastasis to station 4 R). Restaging axial contrast enhanced CT (**i**) and [18]F-FDG PET/CT (**j**) images show increase in size and increase in FDG uptake at right hilar, right mediastinal (4 R) and prevascular nodes, consistent with progression of nodal metastasis. Abnormal nodes were also present at mediastinal 1 R, 2 R and 7 stations post-therapy, which were previously normal at baseline. Subsequent biopsy confirmed carcinoma in the right paratracheal (2 R and 4 R) and subcarinal stations. **k** FNA image of post-ICI abnormal node (station 7 pictured) revealed the presence of malignancy with disease progression (Papanicolaou, x20). Analyses related to the presented images and micrographs were conducted once. NIF, nodal immune flare; CT, computed tomography; FDG, fluorodeoxyglucose; FNA, fine needle aspiration; PET, positron emission tomography; PD, progressive disease.

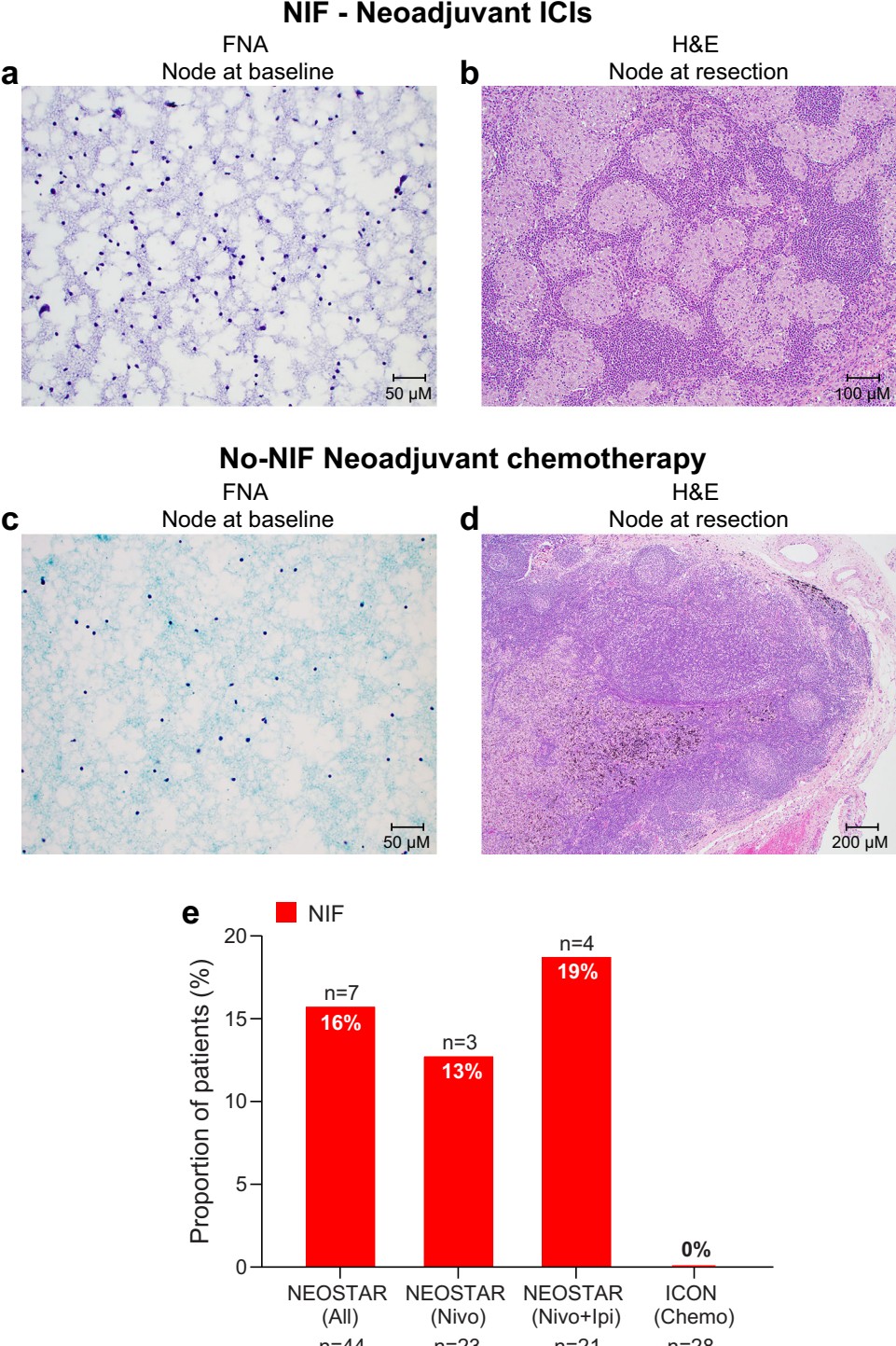

**Fig. 2 Histopathological features of nodal specimens pre- and post-neoadjuvant therapy in NEOSTAR and ICON patients. a** Illustrative FNA image from preoperative mediastinal staging by EBUS in NEOSTAR NIF patient did not demonstrate granulomatous inflammation within examined nodes (station 4 L pictured; Papanicolaou, x20). **b** Resected nodal specimen in NEOSTAR NIF patient following ICIs demonstrating a diffuse non-caseating granulomatous inflammatory reaction (station 11 R pictured; hematoxylin and eosin, x10). **c** Illustrative FNA image from preoperative mediastinal staging by EBUS in ICON No-NIF patient did not demonstrate granulomatous inflammation within examined nodes (station 7 pictured; Papanicolaou, x20). **d** Resected nodal specimen following neoadjuvant chemotherapy in a patient with No-NIF from ICON cohort with the absence of diffuse non-caseating granulomatous inflammatory reaction (station 7 pictured; hematoxylin and eosin, x4). **e** Proportions of patients with NIF, characterized by abnormal nodes on imaging that are devoid of cancer and contain de novo non-caseating granulomas in NEOSTAR (n = 44) and ICON (n = 28) patient cohorts. The red bars depict the proportions of patients with NIF. Analyses related to the presented micrographs were conducted once. NIF, nodal immune flare; ICIs, immune checkpoint inhibitors; ICON, ImmunogenomiC prOfiling in NSCLC; EBUS, endobronchial ultrasound; FNA, fine needle aspiration. Source data for panel (**e**) are provided as a Source Data file.

**Table 2 Changes in tumor volume and SUV$_{max}$ in ICON patients with abnormal nodes post-therapy.**

| ICON cohort | | | | | |
|---|---|---|---|---|---|
| | n | Pre-chemotherapy Median (min, max) | Post-chemotherapy Median (min, max) | Change (Post–Pre) Median (min, max) | P value |
| Tumor volume (mL) | 8[*] | 68.2 (2.50, 207) | 20.6 (2.60, 94.0) | −35.9 (−113, 0.10) | 0.016 |
| Tumor SUV$_{max}$ | 3[**] | 9.30 (7.10, 19.0) | 8.90 (6.50, 9.60) | 0.30 (−14.4, 1.80) | 1.00 |

*ICON* ImmunogenomiC prOfiling in Non-small cell lung cancer. [*]Post-chemotherapy tumor volume was not measurable (atelectasis) in one patient. [**]Six patients did not have paired tumor SUV$_{max}$ values. SUV$_{max}$, maximum standardized uptake value. Two-sided *P* value is from Wilcoxon signed-rank test for paired tumor volume and tumor SUV$_{max}$ comparisons. Source data are provided as a Source Data file.

abnormal nodes post-therapy that were cancer-free with de novo non-caseating granulomas in the final pathological analysis (Fig. 2c, d). Overall, 16% of patients treated with ICIs on the NEOSTAR study (7/44, 95% CI 7–30%) were noted to have NIF (Fig. 2e). Thirteen percent (3/23, 95% CI 3–34%) of cases were observed in the nivolumab monotherapy group, and 19% (4/21, 95% CI 5–42%) were seen in those treated with nivolumab plus ipilimumab (Fig. 2e). No cases of NIF were observed in patients treated with neoadjuvant platinum-based chemotherapy in the ICON cohort (0%, 0/28) (Fig. 2e; Source Data file).

To evaluate the likelihood, by chance, of identifying no cases of NIF in chemotherapy-treated ICON patients, we calculated the probabilities of observing no NIF cases assuming several true incidence rates. The probabilities of observing 0 incidence of NIF in the 28 ICON patients were 23.8%, 5.2%, and only 0.8% if the true NIF rates were 5%, 10%, and 16%, respectively (Supplementary Fig. 1; Source Data file). We further calculated the posterior probability that the NIF rate in the ICON cohort (*p*) was equal to or greater than the observed NIF rate in the NEOSTAR cohort (16%) and a prior p beta (0.16, 0.84), indicating a prior belief that the NIF rate in the ICON cohort was the same as the one in the NEOSTAR cohort. This probability was only 0.04%.

Interestingly, we also noted that an additional 7% (3/44) of patients who were treated with ICIs on trial had nodes with de novo non-caseating granulomas on pathological analysis after treatment but these nodes were not radiologically abnormal post-therapy. The observation of de novo non-caseating granulomas related to all examined lymph nodes, albeit in the absence of nodes suspicious for cancer on imaging post-therapy in some cases, argues for a systemic effect of ICIs on lymph nodes. These findings suggest that the incidence of de novo non-caseating granulomatous nodal inflammation could be as high as 23% (10/44) following at least one cycle and up to three cycles of neoadjuvant ICIs; however, not all cases were associated with false positive radiological evidence of nodal disease progression. Considering these observations, we suggest pathological analysis of any lymph nodes deemed suspicious for disease by standard radiological criteria due to the possibility of false positive findings on restaging imaging.

**Changes in radiological tumor and nodal parameters after neoadjuvant chemotherapy and ICIs.** To determine the changes in the radiologic features of tumor and lymph nodes induced by neoadjuvant chemotherapy and ICIs, we measured the tumor volume and maximum standardized uptake value (SUV$_{max}$), and the nodal size and SUV$_{max}$ in both ICON and NEOSTAR patients with abnormal nodes post-therapy and compared these values to their baseline measurements. In ICON patients, the median tumor volume decreased from 68.2 to 20.6 mL (*P* = 0.016) and the median tumor SUV$_{max}$ decreased from 9.30 to 8.90 after chemotherapy, although this did not reach statistical significance (Table 2; Source Data file). In the same cohort, the mean nodal size decreased from 1.54 cm at baseline (pre-therapy) to 1.25 cm

after chemotherapy (*P* = 0.058) (Fig. 3a; Source Data file), whereas the mean nodal SUV$_{max}$ remained unchanged (Fig. 3b; Source Data file). In NEOSTAR patients with NIF, the mean tumor volume and SUV$_{max}$ did not significantly change after ICIs (Table 3; Source Data file). However, the mean size and SUV$_{max}$ of abnormal nodes increased from 0.91 cm to 1.20 cm (*P* < 0.001; Fig. 3c; Source Data file) and from 2.82 to 6.33 (*P* < 0.001; Fig. 3d; Source Data file), respectively. In NEOSTAR patients in the No-NIF group, the mean tumor volume and SUV$_{max}$ did not significantly change after ICI therapy (Table 3; Source Data file), whereas both the mean size (*P* = 0.013; Fig. 3e; Source Data file) and SUV$_{max}$ (*P* < 0.001; Fig. 3f; Source Data file) of abnormal nodes increased after ICI therapy. While nodal size and SUV$_{max}$ increased in both NIF and No-NIF groups in the NEOSTAR cohort, the magnitude of change of both parameters was greater in patients with NIF as compared to that of patients with No-NIF (mean nodal size difference, *P* = 0.139, Fig. 3g; Source Data file; mean nodal SUV$_{max}$ difference, *P* < 0.001, Fig. 3h; Source Data file). Taken together, these results suggest that nodal size and SUV$_{max}$ can increase after neoadjuvant immunotherapy in both NIF and No-NIF patients, with a greater increase in nodal SUV$_{max}$ in NIF patients, without significant changes in tumor volume and tumor SUV$_{max}$ and illustrate the importance of pathological examination of abnormal nodes on imaging after ICIs.

**Increased immune cell infiltration in nodes of patients with non-caseating granulomas after neoadjuvant immunotherapy.** Next, we questioned whether the composition of the tumor immune infiltrate was associated with the occurrence of NIF after neoadjuvant ICIs. Given the modest number of patients with NIF who also had tissues available for these correlative studies, we included in the NIF group the available samples from all patients with pathological evidence of de novo non-caseating granulomas. First, we analyzed the percentages of tumor cells expressing PD-L1 by immunohistochemistry (IHC) and multiplex immunofluorescence (mIF) staining in available tumor samples resected after ICI therapy from NIF and No-NIF patients. We found no association between NIF and percentages of tumor cells expressing PD-L1 post-therapy by either IHC or mIF staining (Supplementary Table 1 and Supplementary Fig. 2a, b; Source Data file). We then analyzed the immune cell subpopulations in resected tumors from NEOSTAR patients with mIF and found no association between the densities and frequencies of immune subpopulations in the resected tumors and NIF (Supplementary Fig. 2c–k; Source Data file).

To better understand the immune composition of nodes in patients with non-caseating granulomas, we performed gene expression analysis by NanoString of resected nodes from patients with NIF and No-NIF after ICIs (Supplementary Data 1). We found that the immune cell infiltration was significantly greater in the NIF nodes. The cell type scores for immune cells (CD45 or PTPRC$^+$), macrophages, dendritic cells, cytotoxic cells, Th1 cells and exhausted CD8 T cells were greater in nodes from

## ICON - Neoadjuvant chemotherapy

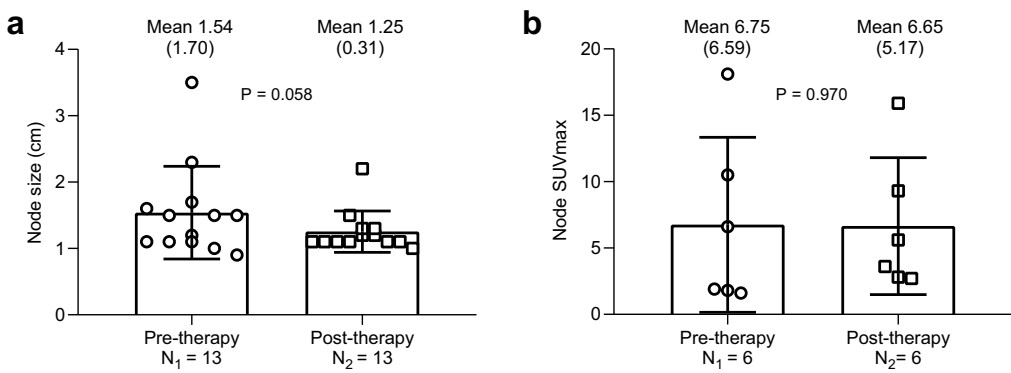

## NEOSTAR - Neoadjuvant ICIs

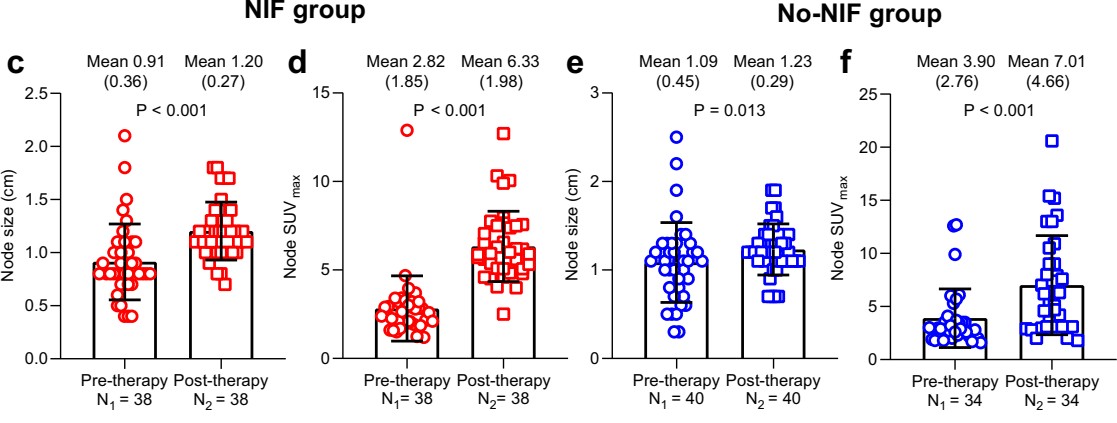

### NIF group

### No-NIF group

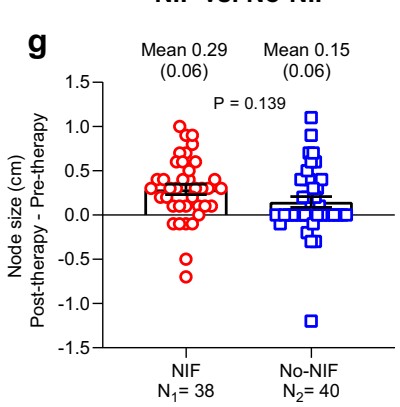

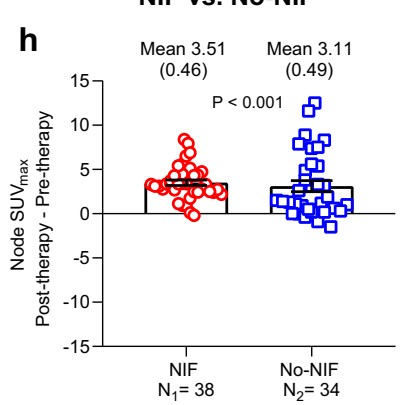

### NIF vs. No-NIF

### NIF vs. No-NIF

patients with NIF than in nodes from patients with No-NIF (Fig. 4a–f, Supplementary Table 2, and Supplementary Data 1; Source Data file). Analysis of differentially expressed genes showed an enrichment of immune-related genes in nodes from patients with NIF as compared to nodes of patients with No-NIF (Fig. 4g; Source Data file). To determine whether the differentially expressed immune genes in nodes of patients with NIF were associated with specific biological processes or molecular functions that may provide insight into the mechanisms governing NIF, we performed gene set enrichment analysis (GSEA) on the resected nodes. We noted that genes usually associated with favorable responses to ICIs, such as the antigen processing and presentation and the interferon gamma (IFN-γ) signaling and response pathways, were significantly upregulated

($P = 0.001$) in nodes from patients with NIF. In contrast, genes involved in immunosuppressive pathways, including the transforming growth factor β (TGFβ) and the SMAD2/3 nuclear signaling pathways were downregulated ($P < 0.005$) in nodes from NIF patients (Fig. 4h; Source Data file). Together, these results suggest that the microenvironment of nodes of patients with NIF/ non-caseating granulomas is inflamed and enriched in macrophages, dendritic and Th1 cells that may be involved in functional pathways of immune response and reduced immune suppression.

**Association of NIF with pathological and radiological tumor responses and ICI-related toxicity.** Since the primary endpoint of NEOSTAR trial was the rate of MPR in resected tumors in the

**Fig. 3 Changes in node size and SUV$_{max}$ in ICON and NEOSTAR patients with abnormal nodes post-therapy. a** Mean node size (cm) of abnormal nodes post-neoadjuvant chemotherapy as compared to pre-therapy in ICON patients. Data are shown as mean node size in cm ±SD. Two-sided *P* value is from linear mixed-effects model. N$_1$ = 13 nodes analyzed in nine patients. N$_2$ = 13 nodes analyzed in nine patients. **b** Mean node SUV$_{max}$ of abnormal nodes post-neoadjuvant chemotherapy as compared to pre-therapy in ICON patients. Data are shown as mean node SUV$_{max}$ ±SD. Two-sided *P* value is from linear mixed-effects model. N$_1$ = 6 nodes analyzed in three patients. N$_2$ = 6 nodes analyzed in three patients. **c**, **d** Mean node size (**c**) and SUV$_{max}$ (**d**) of abnormal nodes post-neoadjuvant ICIs as compared to pre-therapy in NEOSTAR patients with NIF. Data are shown as mean node size in cm ±SD in panel (**c**) and mean SUV$_{max}$ ±SD in panel (**d**). N$_1$ = 38 nodes analyzed in seven patients. N$_2$ = 38 nodes analyzed in seven patients. Two-sided *P* value is from linear mixed-effects model. The red circles and squares depict the node size and SUV$_{max}$ collected from pre-therapy and post-therapy, respectively, in the NIF group. **e**, **f** Mean node size (**e**) and SUV$_{max}$ (**f**) of abnormal nodes post-neoadjuvant ICIs as compared to pre-therapy in NEOSTAR patients with No-NIF. Data are shown as mean node size in cm ±SD in panel (**e**) and mean SUV$_{max}$ ±SD in panel (**f**). N$_1$ = 40 nodes analyzed in 17 patients (**e**); 34 nodes analyzed in 15 patients (**f**) with available scans/images. N$_2$ = 40 nodes analyzed in 17 patients (**e**); 34 nodes analyzed in 15 patients (**f**) with available scans/images. Two-sided *P* value is from linear mixed-effects model. The blue circles and squares depict the node size and SUV$_{max}$ collected from pre-therapy and post-therapy, respectively, in the No-NIF group. **g** Difference in mean size of abnormal nodes between post- and pre-therapy in NEOSTAR patients with NIF as compared with those with No-NIF. Data are shown as change in mean node size in cm ±SE. N$_1$ = 38 nodes analyzed in seven patients. N$_2$ = 40 nodes analyzed in 17 patients. Two-sided *P* value is from linear mixed-effects model. The red circles depict the change of node size in NIF group, and the blue squares depict the change of node size in No-NIF group. **h** Difference in mean SUV$_{max}$ of abnormal nodes between post- and pre-therapy in NEOSTAR patients with NIF as compared with those with No-NIF. Data are shown as change in mean node SUV$_{max}$ ±SE. N$_1$ = 38 nodes analyzed in seven patients. N$_2$ = 34 nodes analyzed in 15 patients. Two-sided *P* value is from linear mixed-effects model. The red circles depict the change of node SUV$_{max}$ in NIF group, and the blue squares depict the change of node SUV$_{max}$ in No-NIF group. ICON, ImmunogenomiC prOfiling in NSCLC; NIF, nodal immune flare; ICIs, immune checkpoint inhibitors; SUV, standardized uptake value; SD, standard deviation; SE, standard error. Source data are provided as a Source Data file.

**Table 3 Changes in tumor volume and SUV$_{max}$ in NEOSTAR patients with abnormal nodes post-therapy.**

**NEOSTAR cohort**

Tumor volume (mL)

| NIF (n = 7) Pre-ICIs | Post-ICIs | P value | No-NIF (n = 18) Pre-ICIs | Post-ICIs | P value | NIF (n = 7) Change (Post–Pre) | No-NIF (n = 18) Change (Post–Pre) | P value |
|---|---|---|---|---|---|---|---|---|
| Mean (SD) | Mean (SD) | | Mean (SD) | Mean (SD) | | Mean (SE) | Mean (SE) | |
| 23.8 (8.60) | 17.1 (13.7) | 0.496 | 49.0 (50.9) | 37.1 (44.5) | 0.059 | −6.67 (9.65) | −12.0 (6.02) | 0.089 |

Tumor SUV$_{max}$

| NIF (n = 7) Pre-ICIs | Post-ICIs | P value | No-NIF (n = 17)ˆ Pre-ICIs | Post-ICIs | P value | NIF (n = 7) Change (Post–Pre) | No-NIF (n = 17)ˆ Change (Post–Pre) | P value |
|---|---|---|---|---|---|---|---|---|
| Mean (SD) | Mean (SD) | | Mean (SD) | Mean (SD) | | Mean (SE) | Mean (SE) | |
| 10.4 (4.40) | 8.98 (6.56) | 0.643 | 14.4 (6.93) | 14.7 (9.05) | 0.867 | −1.39 (2.95) | 0.32 (1.89) | 0.142 |

*n*, Number of patients. *NIF* nodal immune flare, *SD* standard deviation, *SE* standard error. ˆOne patient did not have post-ICI PET/CT images. SUV$_{max}$, maximum standardized uptake value. Two-sided *P* value is from linear mixed-effects model for tumor size and tumor SUV$_{max}$ comparisons. Source data are provided as a Source Data file.

intention-to-treat (ITT) population, we next questioned whether NIF was associated with MPR and/or with the percentage of viable tumor at surgery following neoadjuvant ICIs. We found no difference in MPR rate between NIF (14%, 1/7) and No-NIF groups (32%, 12/37; *P* = 0.357); (Supplementary Table 3). Similarly, we did not observe a distribution difference in the percentage of viable tumor between NIF (median 38%) and No-NIF (median 33%; *P* = 0.842) groups in patients resected on trial (Supplementary Table 4; Source Data file). Also, there were no significant differences between NIF and No-NIF groups in terms of radiological responses evaluable on trial (complete and partial responses, CR + PR) (*P* > 0.99) (Supplementary Table 5) or treatment-related adverse events (TRAEs) (*P* = 0.141) (Supplementary Table 6). The frequencies of TRAEs in NIF and No-NIF patients by grade are shown in Supplementary Tables 7 and 8, respectively. Collectively, these results suggest that the systemic immunological reaction manifested with de novo non-caseating granulomas within the nodes of patients treated with neoadjuvant ICIs may not be associated with therapeutic tumor responses, although larger validation studies are needed.

**Fecal microbiome in patients with NIF and No-NIF.** The fecal microbiome has emerged as an important component influencing cancer responses to ICIs in patients with solid tumors[11]. We analyzed the composition of fecal microbiome between patients with NIF and No-NIF in the NEOSTAR study. Although we did not detect a difference in alpha diversity, which describes the richness of a community, between the groups (Inverse Simpson Index (ISI), *P* = 0.53, Fig. 5a; Source Data file), we did observe a trend in compositional differences between NIF and No-NIF groups as demonstrated by non-overlapping centroids in beta diversity analysis (weighted UniFrac, *r* = 0.20, *P* = 0.06) (Fig. 5b; Source Data file). To avoid inaccurate identification of bacterial taxa at the species level based on the variability within V4 region, which accounts for only about 250 bp of the 1500 bp long 16 S rRNA gene, we restricted the taxonomic classification of the operational taxonomic units (OTUs) to higher taxonomic levels in our analysis stopping at the genus-level classification. In patients with NIF, LDA Effect Size (LEfSe) analysis demonstrated a strong association with genera mostly belonging to the family Coriobacteriaceae of gram-positive phylum Actinobacteria including *Collinsella*, an unclassified genus of Coriobacteriaceae and *Adlercreutzia* (cutoff linear discriminant analysis (LDA) score = 2, *P* = 0.05) (Fig. 5c; Source Data file). These members of Coriobacteriaceae are known to carry out important functions in the intestine such as conversion of bile salts and steroids and the

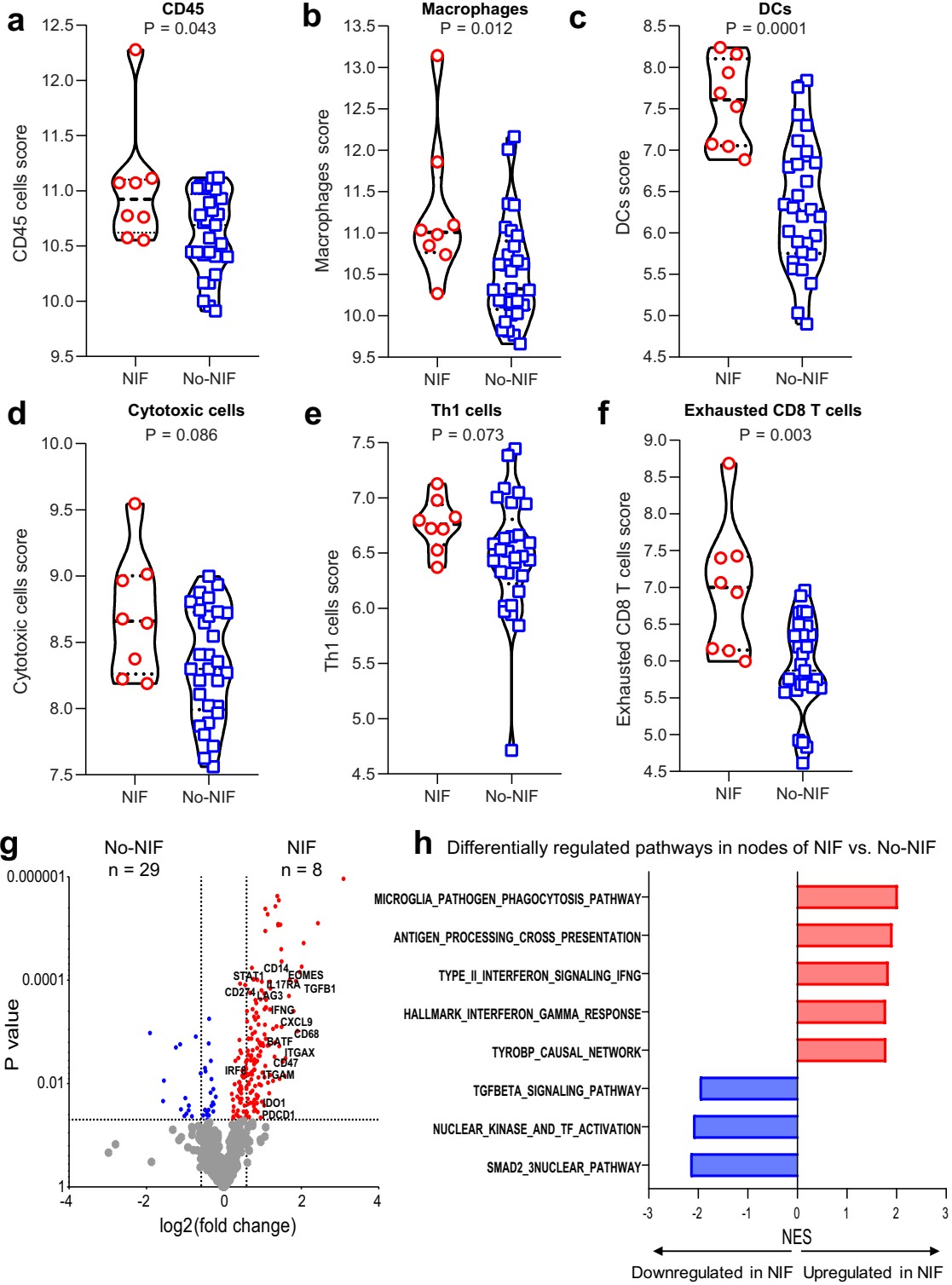

activation of dietary polyphenols. We noted significantly greater abundance of Actinobacteria ($P = 0.044$), Coriobacteriaceae ($P = 0.016$), *Collinsella* ($P = 0.0042$) and *Adlercreutzia* ($P = 0.0064$) in the fecal microbiome of NIF as compared to No-NIF patients (Fig. 5d–g; Source Data file). In contrast, no taxa were estimated to be differentially abundant in No-NIF patients. These exploratory results suggest that the relative abundance of unique fecal microbiome taxa is associated with NIF, a sarcoid-like inflammatory reaction characterized by de novo non-caseating granuloma formation after neoadjuvant ICI treatment.

Subsequent studies will be needed to validate these associations in larger and annotated NIF clinical cohorts.

## Discussion

Herein, we describe a comprehensive report of NIF, a phenomenon that is characterized by abnormal lymph nodes following immune checkpoint therapy in patients with operable NSCLC, mimicking disease progression, only to be classified as cancer-free de novo non-caseating granulomas upon invasive pathological examination. Although sporadic cases of sarcoid-like reaction have been previously

**Fig. 4 Composition of nodal immune infiltrates of NIF/non-caseating granulomas and No-NIF NEOSTAR patients.** NanoString gene expression analysis was performed in tumor-free nodes from patients with NIF ($n = 8$) and No-NIF ($n = 29$). The NIF group for these analyses includes patients with available nodal samples after neoadjuvant therapy that were cancer-free and contained non-caseating granulomas with available nodal NanoString expression data. **a–f** Violin plots show the distribution of immune scores ($\log_2$ normalized counts) in nodes resected from NIF and No-NIF patients in the NEOSTAR study: immune cells expressing CD45 (**a**), macrophages (**b**), dendritic cells (DCs) (**c**), cytotoxic cells (**d**), Th1 cells (**e**), and exhausted CD8 T cells (**f**). The $\log_2$ normalized counts are presented as median with minima, lower and upper quartiles, and maxima. The dashed line indicates the median; the dotted lines indicate the lower quartile and upper quartile values; top and bottom of the violin plots indicate the maxima and minima. The red circles depict data from NIF group, and the blue squares depict data from No-NIF group. **g** Differential expression of genes between NIF and No-NIF nodal samples are illustrated as a volcano plot. Red dots depict significantly upregulated genes in NIF compared to No-NIF nodes and blue dots represent significantly upregulated genes in nodes of No-NIF compared to nodes of NIF patients. **h** Bar plots showing differentially expressed pathways between nodes of NIF and No-NIF patients, computed by GSEA analysis. Red bars indicate pathways that are upregulated while blue bars indicate pathways that are downregulated in nodes of NIF compared to nodes of No-NIF patients. Two-sided $P$ value is from Wilcoxon rank-sum test in panels (**a–f**). Two-sided $P$ values are from Welch's $t$-test in panel (**g**). $P$ values (FDR-adjusted $< 0.2$) are from GSEA algorithm in panel (**h**). NIF, nodal immune flare; DCs, dendritic cells; Th1, T helper cells 1. NES, normalized enrichment score. Source data are provided as a Source Data file and Supplementary Data file (Supplementary Data 1).

reported following chemotherapy[12], our findings indicate that NIF appears to be particularly common following ICI therapy.

The incidence of NIF on restaging scans following neoadjuvant ICIs in our study was 16% (7/44). However, we noted that the presence of pathologic de novo nodal non-caseating granulomatous inflammation occurred in up to 23% of patients in our cohort (10/44). This rather high observed rate of this phenomenon calls for an extra vigilance on restaging of patients undergoing ICI therapy. Each of the seven NIF patients with radiological suspicion of nodal progression underwent careful multidisciplinary deliberation about the next course of the treatment. Three patients underwent additional extensive mediastinal nodal sampling and restaging prior to definitive lung cancer resection. Two of these procedures were done under separate anesthesia. One patient underwent fine needle aspiration (FNA) of a newly abnormal left submandibular node post-therapy, which revealed the absence of cancer and evidence of non-caseating granuloma. After this additional invasive process, the patient eventually declined definitive surgical resection. Three patients underwent definitive planned resection despite the possibility of nodal disease progression on imaging. In all three cases, preoperative surgical planning and judgement estimated that complete tumor and nodal resection was possible. The fact that these patients were enrolled on a clinical trial also contributed to the surgical decision making process as long as local and regional control could be achieved. These results illustrate how the suspicion of disease progression within the lymph nodes on imaging after immunotherapy suggest the need for an additional invasive pathological restaging evaluation and careful surgical judgment about complete resectability in order to avoid erroneous changes in the planned treatment in otherwise operable NSCLC patients.

The mechanisms of action of ICIs have prompted reconsideration of the patterns of radiological responses to therapy[13]. The NIF phenomenon reported here is distinct from tumor pseudo-progression in which a tumor initially progresses and later responds to therapy[6,14]. Here, the lymph nodes alone appeared to progress radiographically, yet the tumor remained stable or became smaller. The nodal enlargement and increased metabolic activity characterizing apparent cancer progression are due to nodal inflammation and pathological features of non-caseating granulomas. As with tumor pseudo-progression, recognizing NIF is critical from a clinical perspective because any misinterpretation of restaging findings as treatment failure may lead clinicians to make inappropriate decisions and avoid potentially curative treatments. The biological mechanisms underlying NIF are unclear and it remains to be determined whether NIF correlates with enhanced antitumor immunity. In previous reports, the pathological assessment of enlarged and avid nodes revealed sarcoid-like changes and elevated expression of

PD-L1 on peripheral immune cells upon discontinuation of ICI therapy in NSCLC[15]. The investigators found that an elevated expression of PD-L1 by peripheral blood mononuclear cells was associated with nodal and skin sarcoid-like reaction after nivolumab was discontinued in a case of unresectable NSCLC[15]. The authors speculated that the increase in peripheral PD-L1 may result from cytokines produced by activated immune cells present in sarcoid lesions and/or in the periphery[15,16]. In the current study, we did not observe greater frequencies of PD-L1 + tumor cells or macrophages within resected tumors of patients with nodal non-caseating granulomas after neoadjuvant ICIs. However, these results may have been influenced by the modest number of samples available for the analysis and should be validated in larger studies. Other studies have shown an association between ICI therapy and the onset of sarcoid-like reactions in patients with advanced stage melanomas, lung adenocarcinomas, and other cancers[8,17]. Non-caseating granulomas are comprised of several cellular types, including macrophages, epithelioid cells, multinucleated giant cells in a core that is circumscribed by Th cells, and B cells[18]. Complement, Th cells, and cytokines such as interleukin (IL)-1, IL-6, and IL-17 as well as IFN-γ have been known to recruit macrophages to sites of granulomatous inflammation[19]. Some evidence also suggests that CD4+ Th1-like cells may contribute to the formation of granulomas, and anti-CTLA-4 therapy has been shown to increase the amounts of peripheral Th17 CD4+ cells that produce the proinflammatory cytokine IL-17[20]. The results of our transcriptomic studies in nodes with non-caseating granulomas after neoadjuvant ICIs revealed greater expression of immune genes and immune cells compared to No-NIF nodes, suggesting a nodal inflammatory reaction that is associated with cellular types involved in pathways of immune activation and reduced immune suppression.

It is noteworthy that we did not detect a significant association between NIF and radiographic and pathological tumor responses, suggesting that the cellular and molecular mechanisms of systemic immunity driving NIF and the antitumor immune responses may be distinct from one another. This hypothesis is further supported by the results of our exploratory analysis of the fecal microbiome. Accumulating evidence suggests that the composition of bacteria residing in the gut may play a key role in determining the efficacy of anticancer therapies, including ICIs[21–23]. Recently, we demonstrated that an increased relative abundance of gut *Ruminococcus* and *Akkermansia* was associated with MPR in a cohort of patients with operable NSCLC receiving neoadjuvant ICIs[9]. Here, we found that NIF was associated mainly with bacterial taxa that belong to Actinobacteria including *Collinsella*. *Collinsella* is a known butyrate producer that provides energy to the intestinal epithelium and modulates immune function to ensure host health[24,25]. More recently, a member of

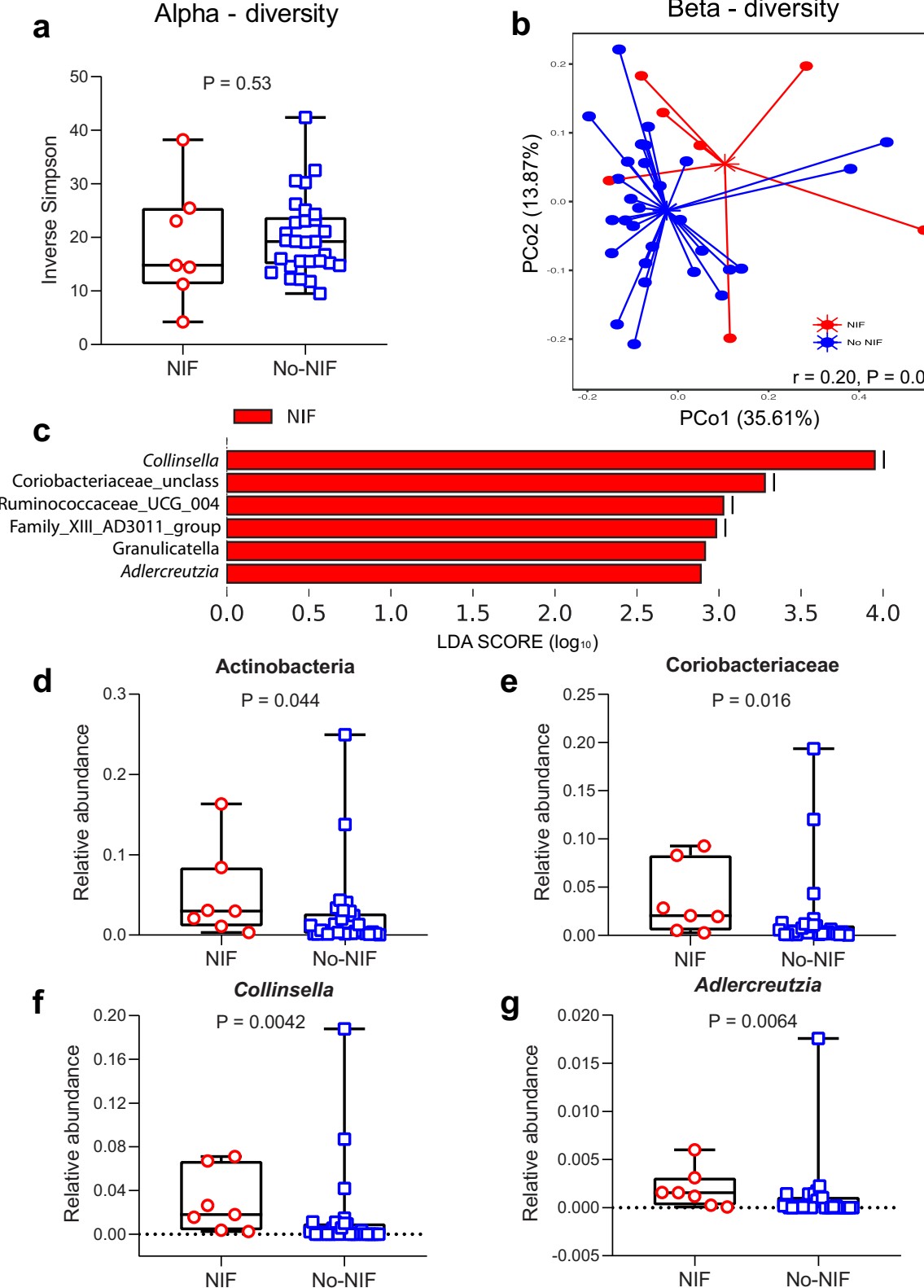

*Collinsella* was shown to be more abundant in metastatic melanoma patients who responded to anti-PD-1 therapy[23] and in patients with low-fiber intake and rheumatoid arthritis—perhaps by influencing the production of inflammatory molecules[26]. The initial associations of relative abundance of unique intestinal microbiota with the NIF phenomenon revealed by our exploratory analyses should serve as a foundation to build upon for subsequent validation in larger clinical NIF datasets and for mechanistic studies aimed to determine the potential functional role of gut microbiota in the formation and composition of nodal non-caseating granulomas in patients developing NIF after neoadjuvant ICIs.

**Fig. 5 Analysis of gut microbiome diversity and composition conducted by sequencing V4 region of 16 S rRNA gene in NIF and No-NIF NEOSTAR patients. a** Inverse Simpson measuring alpha diversity of fecal microbiome in NIF ($n = 7$) and No-NIF ($n = 29$) patients. Data are presented as median with minima, lower and upper quartiles, and maxima. The ends of the box are the upper and lower quartiles (75th and 25th percentiles), the median is the horizontal line inside the box. The whiskers are the two lines outside the box that extend to the maxima and minima. Two-sided $P$ value is from Mann-Whitney $U$ test. The red circles depict data from the NIF group, and the blue squares depict data from the No-NIF group. **b** Ordination plot based on the principal coordinate analysis (PCoA) using weighted UniFrac demonstrating taxonomic similarities between NIF and No-NIF patients. The two axes of the ordination plot explained 49.48% variation in the dataset. Analysis of similarity (ANOSIM) test was used to test whether there is a significant difference between these two groups with 1000 permutations ($r = 0.20$; $P = 0.06$). **c** Linear discriminant analysis (LDA) scores ($\log_{10}$) calculated for differentially abundant bacterial taxa at the genus level in the fecal microbiomes of NIF and No-NIF patients using LDA cutoff of 2 and two-sided $P$ value cutoff of 0.05. **d**–**g** Few most differentially abundant bacterial taxa present in fecal samples of NIF ($n = 7$) and No-NIF ($n = 29$) patients. Relative abundance comparisons of (**d**) Actinobacteria (phylum), (**e**) Coriobacteriaceae (family), (**f**) Collinsella (genus), and (**g**) Adlercreutzia (genus) between NIF and No-NIF patients. Data are presented as median with minima, lower and upper quartiles, and maxima. The ends of the box are the upper and lower quartiles (75th and 25th percentiles), the median is the horizontal line inside the box. The whiskers are the two lines outside the box that extend to the maxima and minima. Two-sided $P$ value is from Mann-Whitney $U$ test. The red circles depict data from the NIF group, and the blue squares depict data from the No-NIF group. Source data are provided as a Source Data file.

While our study sheds light on the NIF phenomenon and its relationship to neoadjuvant ICIs in NSCLC, it does have limitations. The overall sample size is relatively small and with a modest power to detect significant differences in some cases. The small sample size also prohibited more robust mechanistic studies of this phenomenon. Furthermore, the heterogeneity in clinical characteristics between the neoadjuvant chemotherapy- and immunotherapy-treated patient cohorts, e.g., differences in histology, nodal involvement and overall stage distribution, may have influenced the results in these cohorts. Our results will require validation in larger cohorts of patients treated with neoadjuvant ICI-based therapies and it remains to be seen whether the number of cycles and the time from last dose of therapy to restaging imaging may contribute to differences in the systemic immunological response that is manifested as NIF.

In conclusion, surgeons and oncologists administering ICIs as neoadjuvant strategy in NSCLC and nuclear medicine physicians and radiologists reporting their imaging studies need to be attentive to apparent radiological nodal disease progression following neoadjuvant ICIs. Our findings suggest that diligent diagnostic workup, including invasive pathological evaluation where appropriate, of lymph nodes that are deemed suspicious for disease progression by radiological criteria is warranted following neoadjuvant immunotherapy prior to proceeding with final treatment decisions. This report is particularly relevant when considering numerous ongoing neoadjuvant ICI trials and merits further investigation and search for additional tests and biomarkers that may eventually prove helpful in distinguishing NIF from true disease progression without invasive restaging.

## Methods

**Patient cohorts.** NEOSTAR (NCT03158129) is a single-institution, investigator-initiated phase II trial on which the first 44 patients with stage I–IIIA (N2 single station; 7th edition of the American Joint Committee on Cancer, AJCC) operable NSCLC were randomized after baseline computed tomography (CT) and [18]F-FDG positron emission tomography (PET)/CT to receive three doses of neoadjuvant PD-1 inhibitor nivolumab (3 mg/kg intravenously (IV) on days 1, 15, and 29) or combination of nivolumab (3 mg/kg IV on day 1, 15, and 29) plus the CTLA-4 inhibitor ipilimumab (1 mg/kg IV on day 1 only) followed by restaging CT and PET/CT imaging (both recommended at least 14 days after last dose of ICI therapy) and subsequent surgical resection[9]. The primary endpoint of the study was MPR in the resected tumor specimens in the ITT population and has been reported with select secondary and exploratory endpoints[9].

For the current study, we analyzed a cohort of 44 patients treated on the NEOSTAR randomized trial[9]. Among 44 patients, one patient, who experienced TRAE after combination ICIs, was not radiographically evaluable on trial and underwent radiological restaging after neoadjuvant platinum-based chemotherapy administered off trial. Overall, 39 patients underwent surgical resection; among these, 37 patients were resected on trial after ICI therapy, whereas two patients underwent surgery off trial after receiving additional systemic therapies[9]. We also analyzed a comparative cohort of 28 patients who were enrolled on the ImmunogenomiC pROfiling in NSCLC (ICON) project at our institution[10] (IRB approved; PA15-1112; all patients signed informed consent) and were treated with

standard of care neoadjuvant platinum doublet chemotherapy for stage IB–IIIA NSCLC followed by surgical resection.

**Radiological assessment.** CT scans were acquired in a multidetector scanner following IV contrast administration unless contraindicated. Multiplanar CT image series were reconstructed with 2.5 mm slice thickness using standard and high spatial reconstruction algorithms. FDG-PET/CT imaging was performed using Discovery STE PET/CT scanner (GE Healthcare, Waukesha, WI, USA). All patients fasted for 6 h before the FDG injection and had confirmed normal fasting blood glucose level of less than 200 mg/dL. PET was performed in three-dimensional mode at 3–5 min per bed station depending on patient BMI. An intravenous injection of 9–11 mCi of FDG was administered in the arm or central venous catheter on the side opposite to the cancer, and emission scans were acquired at $70 \pm 10$ min after the FDG injection. The acquired PET data were corrected for scatter coincidences, random coincidences, deadtime, and attenuation and reconstructed using OSEM on standard vendor-provided workstations. Non-contrast-enhanced CT images from the base of the skull to the mid-thigh were acquired in helical mode (speed, 13.5 mm per rotation) during shallow breathing at a 3.75 mm slice thickness, a tube voltage of 120 kVp, and 0.5 s rotation with tube current modulation. Daily quality control procedures were performed on all PET scanners to ensure cross-calibration between systems and normalize differences in system performance. In a small number of patients, CT or PET/CT scans were performed at an outside institution with comparable technique. All available CT and PET/CT images were reviewed in all patients in both cohorts by a board-certified thoracic radiologist. Measurements of short-axis diameter on CT and [18]F-FDG maximum standardized uptake value ($SUV_{max}$) were recorded for all abnormal mediastinal or hilar nodes, which were defined as nodal short-axis diameter > 1.0 cm (>1.2 cm in the subcarinal region)[27] on CT images and/or $SUV_{max} \geq 4.5$ on PET/CT images post-therapy and compared to their corresponding pre-therapy measurements. Measurements were also obtained in any abnormal extra-thoracic node if it met CT and [18]FDG uptake inclusion criteria described above. Tumor volume measurement was performed using a commercially available semi-automatic software MIM v.6.6.6 (MIM Software Inc., Cleveland, OH, USA). Characterization of nodal size on CT was performed by measuring short-axis diameter using mediastinal window setting (level 50 HU; width 350 HU). Characterization of tumor and mediastinal lymph node [18]F-FDG uptake was performed using semiquantitative analysis of the $SUV_{max}$ (MIM v.6.6.6; MIM Software Inc., Cleveland, OH, USA). In a few cases, inclusion of outside PET/CT scans or the lack of available images precluded $SUV_{max}$ measurements. As described above, on [18]F-FDG PET/CT a $SUV_{max}$ cutoff value of ≥4.5 was used to characterize abnormal nodes suspicious for malignancy. Although the traditional $SUV_{max}$ cutoff of 2.5 is associated with a lower false negative rate, which is important to avoid missing nodal metastasis, higher $SUV_{max}$ cutoffs have been shown to improve PET/CT diagnostic performance in NSCLC nodal staging by decreasing the false positive rate and may be helpful in distinguishing malignant from inflammatory nodes[28–32]. Positive findings were compared with the post-therapy cytopathologic/histopathological specimen findings and pathological staging for all patients.

For the radiological analysis of the NEOSTAR cohort, all available images in 44 patients, including the patient who underwent restaging imaging off trial after one dose of ICIs followed by administration of platinum doublet chemotherapy off study, were analyzed. The post-therapy size and $SUV_{max}$ measurements of abnormal nodes were recorded and compared to the corresponding pre-therapy nodal size and metabolic activity measurements in 24 and 22 patients, respectively. For the radiological analysis of the ICON cohort, all available images in 28 patients were analyzed. The post-therapy nodal size and $SUV_{max}$ measurements of abnormal nodes were recorded and compared to the corresponding pre-therapy nodal size and metabolic activity measurements in nine and three patients post-neoadjuvant chemotherapy, respectively.

**Pathological analysis.** For this study, the final pathological node analysis to assess the presence of cancer or non-caseating granulomas after neoadjuvant therapy was

performed in all available post-therapy samples from the NEOSTAR cohort ($n = 41$: 37 resected on trial, two resected off trial, and two not resected that had available FNA material of abnormal nodes post-therapy) and in available post-therapy samples in resected patients from the ICON cohort ($n = 22$). Pathological evaluation consisted of the cytopathologic (FNA)/histopathological examination of the sampled/resected lymph nodes post-therapy. Examination of pre-therapy lymph nodes consisted of review of available FNA/histological material obtained during invasive mediastinal staging procedures. Pathological evaluation also consisted of the gross and histopathological examination of the lung resection specimens in the patients resected on trial[9,33,34]. MPR was achieved in tumors with less or equal to 10% viable tumor, and pathological complete response (pCR) was achieved in tumors with 0% viable tumor. Resected mediastinal and peribronchial lymph nodes were submitted and processed for microscopic assessment in a routine fashion and evaluated for the presence of an inflammatory sarcoid-like reaction with non-caseating granulomas.

**Exploratory analyses**. We performed exploratory analyses to study the association of NIF with tumor and nodal immune infiltrate, with the radiological (as determined by Response Evaluation Criteria in Solid Tumors, RECIST v1.1[35]) and pathological (as determined by MPR status in ITT population and percentage viable tumor cells in resected tumor specimens) responses to neoadjuvant ICIs, TRAEs, and fecal microbiome diversity and composition. Given the overall modest number of NIF cases with post-therapy tumor and nodal immune infiltrate data points for the immune correlative studies performed with IHC, mIF and Nano-String, we included in the NIF group of these analyses the data points from all available post-therapy tissue samples with nodal non-caseating granulomas.

**Immunohistochemistry of PD-L1 and analysis**. Available post-therapy formalin-fixed paraffin-embedded (FFPE) tumor tissues were used to evaluate the percentage of malignant cells with membrane PD-L1 expression (clone 28-8, dilution 1:100; Abcam cat# ab205921, Cambridge, MA, USA) by single chromogenic IHC assay. Staining conditions with this anti-PD-L1 clone were optimized and validated[36] by our group using the Leica Bond Max autostainer (Leica Biosystems). The IHC staining was performed in a Leica Bond Max autostainer system according to standard automated protocols. Briefly, tissue sections were deparaffinized and rehydrated following the Leica Bond protocol; antigen retrieval was performed with Bond Solution #2 (Leica Biosystems, equivalent to EDTA, pH 9.0) for 20 min; the primary antibody was incubated for 15 min at room temperature and detected using the Bond Polymer Refine Detection kit (Leica Biosystems) with DAB as chromogen. The slides were counterstained with hematoxylin, dehydrated, and cover-slipped[9]. Microscopy evaluation was performed by two pathologists following the International Association for the Study of Lung Cancer (IASLC) guidelines[37]. Experiments and scoring related to the presented results were conducted once. The immunohistochemical staining and analysis were performed in post-therapy tumor tissues from six NIF and 22 No-NIF patients. Data were collected using Microsoft Excel v.2016 and plotted using GraphPad Prism v.8.0.0.

**Multiplex immunofluorescence staining and analysis**. mIF staining was performed using methods and reagents that have been validated by our group[38]. Briefly, 4 μm thick FFPE tumor sections were stained using an automated staining system (BOND-RX; Leica Microsystems, Buffalo Grove, IL, USA) and two panels containing antibodies against the following markers, Panel 1: cytokeratin (clone AE1/AE3, cat# M351501-2, dilution 1:300, Dako, Santa Clara, CA, USA), PD-L1 (clone E1L3N, cat# 13684 S, dilution 1:3000, Cell Signaling Technology, Danvers, MA, USA), CD68 (clone PG-M1, cat# M087601-2, dilution 1:450, Dako), CD3 (polyclonal, cat# IS503, dilution 1:100, Dako), CD8 (clone C8/144B, cat# MS-457-S, dilution 1:300, Thermo Fisher Scientific, Waltham, MA, USA), and PD-1 (clone EPR4877-2, cat# ab137132, dilution 1:250, Abcam, Cambridge, MA, USA); and Panel 2: cytokeratin (clone AE1/AE3, cat# M351501-2, dilution 1:300, Dako), CD3 (polyclonal, cat# IS503, dilution 1:100, Dako), CD8 (clone C8/144B, cat# MS-457-S, dilution 1:300, Thermo Fisher Scientific), CD45RO (clone UCHL1, cat# PA0146, Cell Signaling Technology), Granzyme B (clone 11F1, cat# PA0291,Cell Signaling Technology), and FOXP3 (clone D2W8E, cat# 98377 S, dilution 1:50, Cell Signaling Technology)[9]. All markers were sequentially applied and stained using their respective fluorophores in the Opal 7 kit (catalogue #NEL797001KT; Akoya Biosciences/PerkinElmer, Waltham, MA, USA)[39]. Stained slides were scanned using the multispectral microscope, Vectra 3.0.3 imaging system (Akoya Biosciences/PerkinElmer), under fluorescence and low magnification at 10x[39]. Following scanning, a pathologist (E.R.P.) selected around five regions of interest (each ROI, 0.3345 mm[2]) per sample to cover around 1.65 mm[2] of tumor tissue using the phenochart 1.0.9 viewer (Akoya Biosciences/PerkinElmer). ROIs were analyzed using the InForm 2.8.2 image analysis software (Akoya Biosciences/PerkinElmer). Marker colocalization was employed to identify the following cellular subsets: malignant cells (AE1/AE3$^+$); PD-L1-expressing malignant cells (AE1/AE3$^+$PD-L1$^+$); T cells (CD3$^+$); cytotoxic T cells (CD3$^+$CD8$^+$); antigen-experienced T cells (CD3$^+$PD-1$^+$); antigen-experienced cytotoxic T cells (CD3$^+$CD8$^+$PD-1$^+$); macrophages (CD68$^+$); PD-L1-expressing macrophages (CD68$^+$PD-L1$^+$); cytotoxic activated T cells (CD3$^+$CD8$^+$Granzyme B$^+$); effector/memory cytotoxic T cells (CD3$^+$CD8$^+$CD45RO$^+$); and regulatory T cells (CD3$^+$CD8$^-$

FOXP3$^+$). Cell densities were averaged for each subset and were finally quantified as number of cells/mm$^2$ in tumor nests and in tumor stroma[40]. Malignant cells and macrophages expressing PD-L1 were expressed also in percentages. Data were consolidated using the R studio 3.5.3 (Phenopter 0.2.2 packet, Akoya Biosciences/PerkinElmer) and SAS 7.1 Enterprise. Experiments and quantification related to the presented results were conducted once. The mIF staining and analysis were performed in post-therapy tumor tissues from four NIF and 21 No-NIF patients. The results were collected using Microsoft Excel v.2016 and plotted using GraphPad Prism v.8.0.0.

**NanoString and gene set enrichment analysis (GSEA)**. Node samples were collected from patients who had signed an informed consent to participate in the NEOSTAR study protocol. FFPE tissue samples were cut into 4 μm thick sections and were shipped to the Immunotherapy Platform at our institution for Nano-String analysis. The analysis was performed as per the umbrella protocol PA13-0291. Tissue sections were de-waxed using deparaffinization solution (Qiagen, Valencia, CA, USA) and total RNA was extracted using the RecoverALL™ Total Nucleic Acid Isolation kit (Ambion, Austin, TX, USA) according to the manufacturer's instructions. RNA quality and quantity were assessed using the Nano-drop Spectrometer (ND-Nanodrop1000, Thermo Fisher Scientific, Wilmington, MA, USA). For the assay, 100 ng of RNA was used to detect immune gene expression using the nCounter PanCancer Immune Profiling panel along with custom CodeSet. nCounter Digital Analyzer was used to tabulate the counts of the reporter probes and for further analysis raw data output was imported into nSolver (http://www.nanostring.com/products/nSolver). Normalization, cell type, and differential gene expression analyses were performed using the nSolver Advanced Data analysis package. GSEA was performed using GSEA software (https://www.gsea-msigdb.org)[41,42]. The nodal transcriptomic analyses were performed in non-cancerous post-therapy nodes from eight NIF and 29 No-NIF patients. Data were collected using Microsoft Excel v.2016 and plotted using GraphPad Prism v.8.0.0.

**Association of NIF with radiographic and pathological responses and TRAEs**. For the analyses evaluating the association of NIF with radiographic responses, MPR, and TRAEs, we included the ITT of 44 NEOSTAR patients. For the analyses evaluating the association of NIF with the percentage viable tumor at resection, we included 37 NEOSTAR patients resected on trial after the administration of ICIs.

**Fecal microbiome sample processing and analysis**. Total DNA was extracted from fecal samples using the QIAamp DNA Stool Kit (Qiagen, Hilden, Germany), including a bead-beating lysis step. The V4 region of bacterial 16 S ribosomal-RNA V4 region was amplified and sequenced on the Illumina MiSeq platform with 2 × 250 bp reads (Illumina, Inc., San Diego, CA, USA). We used VSEARCH v2.10.4 to merge and de-replicate paired-end reads and sorted them by length and size. Sequences were then error-corrected and chimera-filtered using the UNOISE algorithm v.3 and generated OTUs and presumed chimeras. Later, we added the chimera sequences identified by the UNOISE algorithm v.3 but matched an entry in Silva database version 138[43] with a perfect score back to the OTU list and generated a total of 1849 OTUs. The sequencing depths ranged from 1339 to 175,238, with a median of 11,656 reads per sample. Alpha diversity was calculated using Inverse Simpson Index. Weighted-UniFrac dissimilarity indices were used to calculate the pairwise dissimilarities and perform principal coordinate analysis (PCoA) between samples[44]. Alpha and beta diversities were calculated using QIIME 1.9.0[45]. LDA was performed using the LEfSe algorithm[46] for comparing bacterial taxa at the genus level between groups. The microbiome diversity and composition analyses were performed in seven NIF and 29 No-NIF patients. The results were plotted in R (R Core Team 2020; https://www.R-project.org) using ggplot2 package (https://ggplot2.tidyverse.org) and GraphPad Prism v.8.00.

**Statistics**. Patient characteristics at baseline were summarized. Point estimate of the incidence of NIF along with exact 95% confidence interval (CI) were provided. In each patient, multiple lymph nodes were analyzed to estimate changes in nodal size and nodal SUV$_{max}$ following neoadjuvant chemotherapy compared to pre-chemotherapy at baseline in ICON patients and after neoadjuvant ICIs compared to pre-ICIs at baseline in NEOSTAR patients. The Wilcoxon signed-rank test was used for paired comparisons in tumor volume and tumor SUV$_{max}$ in ICON cohort. We utilized the linear mixed-effects model to consider the intra-individual correlation of multiple lymph nodes for the nodal size and nodal SUV$_{max}$ analyses in ICON and NEOSTAR cohorts and for the tumor volume and tumor SUV$_{max}$ analyses in the NEOSTAR cohort. Unconditional exact test available in the R package "Exact" was used to compare the categorical distributions between patient populations. The Wilcoxon rank-sum test was used to compare the distributions of continuous variables and immune markers between patient populations. Two-sided $P$ values < 0.05 were considered significant. Analyses were performed in R 3.6.3 and UNIVARIATE, NPAR1WAY, and MIXED procedures in SAS 9.4. For the differential gene expression analysis, $P$ values for volcano plot were obtained using Welch's $t$-test and determined by nSolver advanced analysis 2.0 software. For the pathway analysis, $P$ values (FDR-adjusted < 0.2) were derived from GSEA algorithm (https://www.gsea-msigdb.org)[41,42]. For the fecal microbiome analysis in

patients with NIF and No-NIF, LDA score of 2 and two-sided $P$ value of 0.05 were used as cutoff. Two-sided $P$ values were from Mann-Whitney $U$ test.

**Ethical approval**. Informed consent was obtained from all study participants. The study was approved by The University of Texas MD Anderson Cancer Center's Institutional Review Board (NEOSTAR, NCT03158129; ImmunogenomiC prOfiling in Non-small cell lung cancer, ICON, PA15-1112; Immunotherapy Platform umbrella protocol PA13-0291). This study complied with all relevant regulations regarding the use of human study participants and was conducted in accordance with the criteria set by the Declaration of Helsinki.

**Reporting summary**. Further information on research design is available in the Nature Research Reporting Summary linked to this article.

## Data availability

The authors declare that the data supporting the findings of this study are available within the manuscript, its supplementary information files, and the Source Data. The 16S fecal microbiome sequencing data (supporting the findings in Fig. 5) are publicly available in the National Center for Biotechnology Information Sequence Read Archive (SRA BioProject ID PRJNA665109). Taxonomy was assigned using the Silva database (https://www.arb-silva.de/) for 16 S rRNA sequences. The NanoString $\log_2$ normalized counts data that support the findings of this study (Fig. 4 and Supplementary Table 2) are available as Supplementary Data 1 with the manuscript. Other relevant de-identified data/information related to the current study that can be shared will be available from the corresponding authors (T.C. and J.V.H.) at earliest convenience and within a reasonable timeframe upon reasonable academic request and will require the researcher to sign a data access agreement with the University of Texas MD Anderson Cancer Center as the information includes data collected under an institutional alliance clinical trial and/or an institutional observational protocol. Source data are provided with this paper.

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

## Acknowledgements

We thank the patients and their families who participated in the reported research, and all members of our clinical trial, data, and regulatory research teams for their assistance. The authors would like to thank the members of the strategic alliance teams at Bristol Myers Squibb and MD Anderson Cancer Center for their support; the MD Anderson Cancer Center's Program for Innovative Microbiome and Translational Research (PRIME-TR), which supported the analysis and interpretation of the microbiome results presented herein (Drs. Wargo and Ajami are the program director and executive scientific director for PRIME-TR, respectively); Ashura Khan (Program Director), Marla Polk (Administrative Director) for logistic support and other members of the Immunotherapy Platform at MD Anderson Cancer Center for their technical support and scientific input (Drs. Allison and Sharma are the executive director and scientific director of the Immunotherapy Platform, respectively); B. Sanchez Espiridon and S. Wijeratne for their assistance with sample procurement and inventory and other members of the MD Anderson Cancer Center's Translational Molecular Pathology Immunoprofiling Laboratory (TMP-IL) for their technical support. Support for the study was partially provided by Bristol Myers Squibb, the National Institutes of Health (NIH)/National Cancer Institute (NCI) through The University of Texas Lung Specialized Program of Research Excellence (SPORE) grant P50CA070907 (to T.C., I.I.W., D.L.G., and J.V.H.), the NIH/NCI P30 CA016672 Cancer Center Support Grant, the Conquer Cancer Foundation of the American Society of Clinical Oncology Career Development Award 2018 Project ID 12895 (to T.C.), the Bruton Endowed Chair in Tumor Biology (to J.V.H.), and the TMP-IL at the Department of Translational Molecular Pathology, the University of Texas MD Anderson Cancer Center. The study was also partially supported by the generous philanthropic contributions to the University of Texas MD Anderson Cancer Center Lung Cancer Moon Shot Program, Physician Scientist Program (to T.C.), the Khalifa Scholar Program Award (from the Khalifa Bin Zayed Al Nahyan Foundation, to T.C.), Dr. and Mrs. Weaver (The Gil and Dody Weaver Foundation), the Rexanna's Foundation for Fighting Lung Cancer (to T.C., D.L.G., J.V.H., and B.S.) and the Bob Mayberry Foundation (to T.C.).

## Author contributions

T.C., A.A.V., J.V.H., and B.S. conceived the project, designed the analyses, interpreted the data and wrote the manuscript. T.C. and B.S. collected the clinical data, supervised and participated in data analysis at all stages, and wrote the manuscript. K.G.M. contributed to the collection of radiological and clinical data. A.W. and A.P. collected and analyzed the pathologic data. M.C.B.G. and B.S. collected, analyzed, and interpreted the radiological data. C.H.L., H.Y.L., and J.J.L. designed the statistical plan, performed the statistical analyses, and participated in writing the manuscript. M.A.W.K. and Y.S. performed the microbiome data analysis and participated in writing the manuscript. P.S. and J.P.A. supervised the Immunotherapy Platform team, which performed NanoString gene expression assay of lymph node samples, and provided interpretation of the NanoString data and participated in writing the manuscript. S.B. and S.S.Y. are members of the Immunotherapy Platform and participated in data interpretation and writing of the manuscript. L.M.S. and E.R.P. performed the tumor immune marker staining analyses and participated in writing the manuscript. C.H. supervised the staining analyses. C.H. and H.K. participated in the interpretation of the immune marker staining findings and in reviewing the manuscript. I.I.W. supervised the tumor immune marker analyses at the TMP-IL. N.J.A., J.A.W., R.R.J., and B.S. supervised and participated in the microbiome data analysis and interpretation and in writing the manuscript. W.N.W.Jr. participated in conceiving the project, interpreting the results, and reviewing the manuscript. D.L.G. and S.G.S. participated in writing the manuscript. All authors reviewed the manuscript at all stages.

## Competing interests

T. Cascone has received speaker's fees from the Society for Immunotherapy of Cancer, Bristol Myers Squibb and Roche; reports consultant/advisory role fees from MedImmune/AstraZeneca, Bristol Myers Squibb, EMD Serono, Merck & Co., Genentech and Arrowhead Pharmaceuticals; and reports clinical research funding to The University of Texas MD Anderson Cancer Center from Boehringer Ingelheim, MedImmune/AstraZeneca, Bristol Myers Squibb, and EMD Serono. M.C.B. Godoy has received research funding from Siemens Healthcare. W.N. William Jr. has received honoraria/speaker's fees and/or participated in advisory boards from Roche/Genentech, Bristol Myers Squibb, Eli Lilly, Merck, AstraZeneca, and Pfizer. C. Haymaker reports personal fees from Nanobiotix and science advisory board fees from Briacell. H. Kadara has received funding from Johnson and Johnson. I.I. Wistuba has provided consulting or advisory roles for AstraZeneca/MedImmune, Asuragen, Bayer, Bristol Myers Squibb, Genentech/Roche, GlaxoSmithKline, Guardant Health, HTG Molecular Diagnostics, Merck, MSD Oncology, OncoCyte, Novartis, Flame Inc., and Pfizer; has received grants and personal fees from Asuragen, Genentech/Roche, Bristol Myers Squibb, AstraZeneca/MedImmune, HTG Molecular, Merck, and Guardant Health; has received personal fees from GlaxoSmithKline, OncoCyte, Daiichi-Sankyo, Roche, AstraZeneca, Pfizer, and Bayer; has received research funding to his institution from 4D Molecular Therapeutics, Adaptimmune, Adaptive Biotechnologies, Akoya Biosciences, Amgen, Bayer, EMD Serono, Genentech, Guardant Health, HTG Molecular Diagnostics, Iovance Biotherapeutics, Johnson & Johnson, Karus Therapeutics, MedImmune, Merck, Novartis, OncoPlex Diagnostics, Pfizer, Silicon Biosystems, Takeda, and Novartis. P. Sharma reports consulting, advisory roles, and/or stocks/ownership for Achelois, Apricity Health, BioAlta, Codiak BioSciences, Constellation, Dragonfly Therapeutics, Forty-Seven Inc., Hummingbird, ImaginAb, Jounce Therapeutics, Lava Therapeutics, Lytix Biopharma, Marker Therapeutics, Oncolytics, Infinity Pharma, BioNTech, Adaptive Biotechnologies, and Polaris; and owns a patent licensed to Jounce Therapeutics (61/247,438; "Combination Immunotherapy for the Treatment of Cancer"). J.P. Allison reports consulting, advisory roles, and/or stocks/ownership for Achelois, Apricity Health, BioAtla, Codiak BioSciences, Dragonfly Therapeutics, Forty-Seven Inc., Hummingbird, ImaginAb, Jounce Therapeutics, Lava Therapeutics, Lytix Biopharma, Marker Therapeutics, Polaris, BioNTech, and Adaptive Biotechnologies; and owns a patent licensed to Jounce Therapeutics (61/247,438; "Combination Immunotherapy for the Treatment of Cancer"). J.A. Wargo is an inventor on a US patent application PCT/US17/53,717, "Methods for enhancing immune checkpoint blockade therapy by modulating the microbiome", submitted by the University of Texas MD Anderson Cancer Center and on a patent "Targeting B Cells To Enhance Response To Immune Checkpoint Blockade" UTSC.P1412US.P1 - MDA19-023. J.A. Wargo reports honoraria from Imedex, Dava Oncology, Omniprex, Illumina, Gilead, PeerView, Physician Education Resource, MedImmune, and Bristol-Myers Squibb. J. Wargo serves as a consultant/advisory board member for Roche/Genentech, Novartis, AstraZeneca, GlaxoSmithKline, Bristol-Myers Squibb, Merck, Diversigen, Micronoma, and Ella Therapeutics. R.R. Jenq receives consultant role fees from Merck, Karius and Microbiome DX, advisory member role fees from Seres and Kaleido and patent licensing fees from Seres (US20170258854A1; "Intestinal microbiota and gvhd"). D.L. Gibbons has served on scientific advisory committees for AstraZeneca, GlaxoSmithKline, Sanofi, Eli Lilly, and Janssen; has received research support from Janssen, Takeda, Ribon Therapeutics, Astellas, and AstraZeneca. S.G. Swisher has participated in advisory committees for Ethicon and for the Peter MacCallum Cancer Center. J.V. Heymach has received research support from AstraZeneca, GlaxoSmithKline, and Spectrum; participated in advisory committees for AstraZeneca, Boehringer Ingelheim, Catalyst, Genentech, GlaxoSmithKline, Guardant Health, Foundation Medicine, Hengrui Therapeutics, Eli Lilly, Novartis, Specrtum, EMD Serono, Sanofi, Takeda, Mirati Therapeutics, Bristol Myers Squibb, BrightPath Biotherapeutics, Janssen Global Services, Nexus Health System, Pneuma Respiratory, Kairos Venture Investments, Roche, and Leads Biolabs; and received royalties and/or licensing fees from Spectrum. B. Sepesi receives consultant/advisory role fees from Bristol Myers Squibb. No potential conflicts of interest are disclosed by the other authors.

## Additional information

[1]Department of Thoracic/Head and Neck Medical Oncology, The University of Texas MD Anderson Cancer Center, Houston, TX, USA. [2]Department of Pathology, The University of Texas MD Anderson Cancer Center, Houston, TX, USA. [3]Department of Thoracic and Cardiovascular Surgery, The University of Texas MD Anderson Cancer Center, Houston, TX, USA. [4]Department of Thoracic Imaging, The University of Texas MD Anderson Cancer Center, Houston, TX, USA. [5]Hospital BP, a Beneficencia Portuguesa de Sao Paulo, Sao Paulo, Brazil. [6]Department of Biostatistics, The University of Texas MD Anderson Cancer Center, Houston, TX, USA. [7]The Immunotherapy Platform, The University of Texas MD Anderson Cancer Center, Houston, TX, USA. [8]Department of Surgical Oncology, The University of Texas MD Anderson Cancer Center, Houston, TX, USA. [9]Department of Statistics, The University of Missouri, Columbia, MO, USA. [10]Department of Translational Molecular Pathology, The University of Texas MD Anderson Cancer Center, Houston, TX, USA. [11]Department of Genitourinary Medical Oncology, The University of Texas MD Anderson Cancer Center, Houston, TX, USA. [12]Department of Immunology, The University of Texas MD Anderson Cancer Center, Houston, TX, USA. [13]Department of Genomic Medicine, The University of Texas MD Anderson Cancer Center, Houston, TX, USA. [14]Cancer Prevention and Research Institute of Texas Scholar in Cancer Research, Houston, TX, USA. [15]These authors contributed equally: John V. Heymach, Boris Sepesi. ✉email: tcascone@mdanderson.org; jheymach@mdanderson.org

