## [Peer Review File. · Nature Communications]

Reviewers' Comments:

Reviewer #1:

Remarks to the Author:

Interesting original paper describing nodal immune-flare features in resectable NSCLC patients receiving neoadjuvant immune-checkpoint inhibitors. The background is clear, the main aim of the study clearly stated in the intro, and the methodology adequate to test the study hypothesis. Although limited by small sample size, data presented are interesting since provide the first systematic description of this IO related emerging phenomena, including potential correlations with treatment outcomes and microbiota. The decision to include neoadjuvant chemo patients as control cohort further increase the quality of the paper, which may be suitable for publication, pending the following minor revisions:

- The discussion looks too long to the reader. Please try to summarize the text highlighting the most relevant findings while avoiding repetitions and redundant sentences.
- Consider to include also clinical characteristics heterogeneity between IO and Chemo cohorts among the potential study limitations.
- In light of the small sample size the results of this work need to be considered preliminary and required to be confirmed in prospective studies including larger cohorts of patients. This concept should be clearly stated in the text while, in my opinion, authors should limit to "suggest" (not recommend) invasive node evaluation throughout the text as well as in the abstract conclusion.
- Some typos in text to be edited.

Reviewer #2:

Remarks to the Author:

Neo-adjuvant represents the future of limited NSCLC and the NEOSTAR study is one of the first trial with positive results. Nodal immune flare is a novel phenomenon in this setting.

Despite having access to unique patient data and samples this study would benefit from important correlative studies with respect to the immune infiltration (IHC: CD3, CD8, Treg.. or RNAseq) within the tumor microenvironment and the incidence of NIF.

In addition, it would be important to present the PD-L1 status of the patients and its impact on NIF incidence as well as tumor mutational burden if available.

Finally, several papers demonstrated the correlation between the microbiome and the immune infiltration (Wargo et al. Science 2018) and this association between immune infiltration-NIF and microbiome would be extremely valuable to further characterize NIF and demonstrate a true cause-effect relationship and not just statistical differences.

Minor:

1. Fig 1 and Table 2 are difficult to interpret. Classic case of NIF and no NIF should be represented in fig 1 as well as histogram to demonstrate the most important result of the trial 0% vs 13% and 19% NIF in IO neo-adjuvant group.
2. The paragraph on probabilities of true vs false NIF is extremely confusing with no table or explanatory figure.
3. Figure 2 would benefit from a different color codes or shape for each category to make it easier to read

Reviewer #3:

Remarks to the Author:

This is an exploratory analysis examining the association of nodal immune flare and response. The methods used are both appropriate and well described. I do however have some comments on the methods used

a) The authors claim to use Fisher's exact test to test categorical variables although (i) I could not identify where this was used and (ii) many biostatisticians (strongly) recommend against its use (except perhaps in very specialised cases) - D'Agostino et al, Amer Stat 1987, Lydersen et al Stat in Med, 2009; Berkson, J Stat Plan & Infer 1978 to quote just a few.

b) The Wilcoxon test DOES NOT test medians - I'm surprised that the statistical authors allowed this approach. Again there are numerous examples where medians are equal and the Wilcoxon test is significant [because it tests a shift in an (undefined) location of the two distributions.

A simple example

Group A: 0 2 7 Group B: 1 2 15 Median Group A/B 3

Freq 20 20 20 20 20 Rank-sum p-val 0.036

Use of these methods & statements of median equality are misleading & tend to perpetuate poor statistical practice.

Reviewer #4:

Remarks to the Author:

The manuscript aims to assess the role of NIF evaluated by using molecular imaging such as PET/CT with FDG in a small cohort of patients undergoing ICIs neoadjuvant therapy for NSCLC. The manuscript is interesting, being the evaluation of response to therapy with ICIs a very hot topic, but from the imaging point of view it appears very poor.

Please below some comments:

1) The rate of pseudo progression in NSCLC from ICIs therapy should be described in the Introduction section rather than in the discussion part.

2) the assumption of SUVmax equal to 4.5 in the Introduction paragraph should be justified or enriched with a reference.

3) The study by Cascone et al in the methods section should be reported as reference

4) the final number of enrolled patients is unclear. The sentences of page 7 lines 122-127 are very confused. I am not able to understand if the enrolled patients are 37 or 35 or 36. Please specify.

5) the final population for the chemotherapy arm is 22 and not 28, as reported in page 7, differently from the result paragraph

6) data about CT and PET interpretation and also acquisition protocol and timing from the ICIs administration is completely missing. Please add some additional sentences. The time and the number of cycles can significantly affect the interpretation of the molecular imaging.

7) for the interpretation of metabolic lymph nodes, the authors used a very old paper (2006) that considered only a PET scanner and not a hybrid scanner. Therefore, the authors should use more appropriate and recent criteria, please check the following papers (PMID: 22459646 or 32525121).

8) results section is complete, although the first paragraph is very hard to read, mainly for the statistical assumptions.

9) the authors should underline that the imaging can miss the presence of NIF in 7% of cases, but it cannot alter the final effect of the ICIs treatment.

10) please discuss more in details the information provide in the page 18 lines 394-397. Is the NIF correlated with an under expression of PDL-1? this concept is not clear.

Tables and figures are OK.

Reviewer #1 (Remarks to the Author): with expertise in lung cancer immunotherapy

Interesting original paper describing nodal immune-flare features in resectable NSCLC patients receiving neoadjuvant immune-checkpoint inhibitors. The background is clear, the main aim of the study clearly stated in the intro, and the methodology adequate to test the study hypothesis. Although limited by small sample size, data presented are interesting since provide the first systematic description of this IO related emerging phenomena, including potential correlations with treatment outcomes and microbiota. The decision to include neoadjuvant chemo patients as control cohort further increase the quality of the paper, which may be suitable for publication, pending the following minor revisions:

Comment:

The discussion looks too long to the reader. Please try to summarize the text highlighting the most relevant findings while avoiding repetitions and redundant sentences.

Response:

Thank you for pointing this out. We have revised the Discussion to remove repetitive sentences and summarize the text to enhance the focus on key findings (**pages 21-24 of revised manuscript**).

Comment:

Consider to include also clinical characteristics heterogeneity between IO and Chemo cohorts among the potential study limitations.

Response:

We have modified the text of the revised manuscript to incorporate this limitation, as detailed below:

Discussion (page 24):

“Furthermore, heterogeneity in clinical characteristics between the neoadjuvant chemotherapy- and immunotherapy-treated patient cohorts, e.g. differences in histology, nodal involvement and overall stage distribution, may have influenced the results in these cohorts.”

Comment:

In light of the small sample size the results of this work need to be considered preliminary and required to be confirmed in prospective studies including larger cohorts of patients. This concept should be clearly stated in the text while, in my opinion, authors should limit to “suggest” (not recommend) invasive node evaluation throughout the text as well as in the abstract conclusion.

Response:

In response to the reviewer’s point, we have removed the word “recommend” when referring to invasive evaluation of abnormal nodes post neoadjuvant immunotherapy and suggested pathological evaluation of abnormal nodes in the revised manuscript. We have also acknowledged that our results require validation in larger patient cohorts, as detailed below:

Abstract (page 4):

“Our findings suggest that apparent radiological progression of nodes may occur due to an inflammatory response after neoadjuvant immunotherapy, and that such cases should be evaluated by pathological examination to distinguish between NIF and true nodal progression and ensure appropriate clinical treatment planning.”

Results (page 17):

“Considering these observations, we suggest pathological analysis of any lymph nodes deemed suspicious for disease by standard radiological criteria due to the possibility of false positive findings on restaging imaging.”

Discussion (page 24):

“Our findings suggest that diligent diagnostic workup, including invasive pathological evaluation where appropriate, of any lymph node that is deemed suspicious for disease progression by radiological criteria is warranted following neoadjuvant immunotherapy prior to proceeding with final treatment decisions.”

Discussion (page 22):

“These results illustrate how the suspicion of disease progression within the lymph nodes on imaging after immunotherapy suggest the need for an additional invasive pathological restaging evaluation and careful surgical judgment about complete resectability in order to avoid erroneous changes in the planned treatment in otherwise operable NSCLC patients.”

Discussion (page 24):

“Our results will require validation in larger cohorts of patients treated with neoadjuvant ICI-based therapies and it remains to be seen whether the number of cycles and the time from last dose of therapy to restaging imaging may contribute to differences in the systemic immunological response that is manifested as NIF.”

Comment:

Some typos in text to be edited.

Response:

Thank you for pointing this out. We have corrected the typos in the revised manuscript.

Reviewer #2 (Remarks to the Author): with expertise in cancer immunotherapy and microbiome

Neo-adjuvant represents the future of limited NSCLC and the NEOSTAR study is one of the first trial with positive results. Nodal immune flare is a novel phenomenon in this setting.

Comment:

Despite having access to unique patient data and samples this study would benefit from important correlative studies with respect to the immune infiltration (IHC: CD3, CD8, Treg.. or RNAseq) within the tumor microenvironment and the incidence of NIF.

Response:

To address this concern, we performed correlative studies to evaluate the association between immune cell subpopulations in resected tumors and post-therapy lymph nodes with the occurrence of NIF in patients that had been treated with ICI therapy on the NEOSTAR trial.

Given the overall modest number of NIF cases with post-therapy tumor and nodal tissues available for the immune correlative studies, we included in the NIF group of these analyses the datapoints

from all available post-therapy tissue samples with nodal non-caseating granulomas. We quantified the immune infiltrates of resected tumors after neoadjuvant ICIs using immunohistochemistry (IHC) and multiplex immunofluorescence (mIF) staining.

We found no differences in the percentages of tumor cells expressing PD-L1 as assessed by IHC and mIF staining in tumor tissues resected after neoadjuvant therapy in NIF as compared with No-NIF patients. We also noted that there was no significant association between NIF and the densities and percent expression of several immune cell subpopulations (total CD3+ T cells, cytotoxic T cells, antigen experienced T cells, macrophages, macrophages expressing PD-L1, activated T cells, regulatory T cells, and memory T cells) within the resected tumors post-therapy as determined by mIF staining. We then focused on the post-ICI therapy nodes from NEOSTAR patients and performed NanoString analysis of tumor-free lymph nodes resected from NIF and no-NIF patients. We found a significant increase in immune genes and immune cell infiltration in NIF nodes as compared to No-NIF nodes. The immune scores of CD45+ cells, macrophages, dendritic cells, cytotoxic cells, Th1 cells and exhausted CD8 T cells were significantly increased in NIF nodes as compared to No-NIF nodes (Fig. 4a-f). Analysis of differentially expressed genes showed an enrichment of immune-related genes in NIF nodes compared to No-NIF nodes. When we performed gene set enrichment analysis (GSEA), we noted that genes involved in favorable responses to ICI therapy (e.g., antigen processing/ presentation, interferon gamma signaling and response pathways) were significantly upregulated in NIF nodes, while genes typically associated with immunosuppressive pathways, (TGF β , SMAD2/3) were downregulated in NIF nodes. Together, these results suggest that the microenvironment of NIF nodes is inflamed and enriched in macrophages, dendritic and Th1 cells that may be involved in functional pathways of immune response and reduced immune suppression.

We have described and discussed these results in the revised manuscript as detailed below:

Results (pages 18-19)

“Increased immune cell infiltration in nodes of patients with non-caseating granulomas after neoadjuvant immunotherapy

Next, we questioned whether the composition of the tumor immune infiltrate was associated with the occurrence of NIF after neoadjuvant ICIs. Given the small number of patients with NIF who also had tissues available for these correlative studies, we included in the NIF group the available samples from all patients with pathological evidence of *de novo* non-caseating granulomas. First, we analyzed the percentages of tumor cells expressing PD-L1 by immunohistochemistry (IHC) and multiplex immunofluorescence (mIF) staining in available tumor samples resected after ICI therapy from NIF and No-NIF patients. We found no association between NIF and percentages of tumor cells expressing PD-L1 post-therapy by either IHC or mIF staining (**Supplementary Table 1; Supplementary Fig. 2a-b**). We then analyzed the immune cell subpopulations in resected tumors from NEOSTAR patients with mIF and found no association between the densities and frequencies of immune subpopulations in the resected tumors and NIF (**Supplementary Fig. 2c-k**).

To better understand the immune composition of nodes in patients with non-caseating granulomas, we performed gene expression analysis by NanoString of resected nodes from patients with NIF and No-NIF after ICIs. We found that the immune cell infiltration was significantly greater in the NIF nodes. The cell type score for immune cells (CD45 or PTPRC⁺), macrophages, dendritic cells, cytotoxic cells, Th1 cells and exhausted CD8 T cells were greater in nodes from patients with NIF than in nodes from patients with No-NIF (**Fig. 4a-f, Supplementary Table 2**). Analysis of differentially expressed genes showed an enrichment of immune-related genes in nodes from patients with NIF as compared to nodes of patients with No-NIF (**Fig. 4g**). To determine whether the differentially-expressed immune genes in nodes of patients with NIF were associated with

specific biological processes or molecular functions that may provide insight into the mechanisms governing NIF, we performed gene set enrichment analysis (GSEA) on the resected nodes. We noted that genes usually associated with favorable responses to ICIs, such as the antigen processing and presentation and the interferon gamma (IFN γ) signaling and response pathways, were significantly upregulated ($P = 0.001$) in nodes from patients with NIF. In contrast, genes involved in immunosuppressive pathways, including the transforming growth factor β (TGF β) and the SMAD2/3 nuclear signaling pathways were downregulated ($P < 0.005$) in nodes from NIF patients (Fig. 4h).”

Discussion (page 22):

“In the current study, we did not observe greater frequencies of PD-L1+ tumor cells or macrophages within resected tumors of patients with nodal non-caseating granulomas after neoadjuvant ICIs. However, these results may have been influenced by the modest number of samples available for analysis and should be validated in larger studies.”

Discussion (page 23):

“The results of our transcriptomic studies in nodes with non-caseating granulomas after neoadjuvant ICIs revealed greater expression of immune genes and immune cells compared to No-NIF nodes, suggesting a nodal inflammatory response that is associated with cellular types involved in pathways of immune activation and reduced immune suppression.”

Comment:

In addition, it would be important to present the PD-L1 status of the patients and its impact on NIF incidence as well as tumor mutational burden if available.

Response:

We included the number of NEOSTAR patients with post-therapy tumor samples that were assessed retrospectively by chromogenic immunohistochemistry (IHC) assay to determine the percentage of malignant cells expressing PD-L1, and by multiplex immunofluorescence (mIF) staining assay to assess the percentage of malignant cells and macrophages expressing PD-L1 on available tumor (total tumor and stromal components) tissue samples. Tumor cell PD-L1 expression status and the percentage of malignant cells expressing PD-L1 were assessed in six tumors resected from patients with NIF/non-caseating granulomas and in 22 tumors resected from No-NIF patients by IHC (Supplementary Table 1; Supplementary Fig. 2a). The frequencies of malignant cells and macrophages expressing PD-L1 were assessed in four tumors resected from patients with NIF/non-caseating granulomas and in 21 tumors resected from No-NIF patients by mIF staining (Supplementary Fig 2b and 2h).

As discussed above, we found no differences in tumor cell PD-L1 expression status and the percentage of malignant cells and macrophages expressing PD-L1 within the resected tumor tissues from NIF patients as compared to those from No-NIF patients. It should be noted that our analysis was performed on a small number of patients, and thus larger studies are needed to clarify the impact of tumor PD-L1 expression on NIF.

The results of these analyses are shown in **Supplementary Table 1, Supplementary Fig. 2a-b (PD-L1); Supplementary Fig. 2c-k (immune populations)**. We described these findings in the **Result section (page 18)** and discussed them in the **Discussion section (page 22)** of the revised manuscript.

Comment:

Finally, several papers demonstrated the correlation between the microbiome and the immune infiltration (Wargo et al. Science 2018) and this association between immune infiltration-NIF and microbiome would be extremely valuable to further characterize NIF and demonstrate a true cause-effect relationship and not just statistical differences.

Response:

We appreciate the reviewer’s stance on characterizing microbe-host interactions beyond statistical associations. Although the association between NIF and neoadjuvant immunotherapy and, in turn, of NIF with specific microbial taxa, such as *Collinsella* and *Adlercreutzia*, remains valid based on our studies, further analyses were limited by the very modest number of samples available with both gut microbiome metrics and tumor immune infiltrate datapoints to carry out the mechanistic studies necessary to demonstrate a true cause-effect relationship suggested by the reviewer. Nonetheless, we carried hypothesis-generating analyses by comparing gut microbiome metrics such as diversity, measured by Inverse Simpson Index (ISI), and the relative abundance of *Collinsella* and *Adlercreutzia*, which were found to be statistically associated with NIF (Figure 5 of revised manuscript), with tumor and nodal immune profiles.

We performed Spearman correlations between the ISI, the relative abundance of *Collinsella* and *Adlercreutzia*, and the tumor immune markers assessed by mIF staining in four patient samples with NIF/noncaseating granulomas (NIF group) with available both gut microbiome and immune marker datapoints. The results, although only valid to generate testable hypotheses in subsequent and larger datasets, showed no significant associations between the diversity and relative abundance of these bacteria with the frequencies of malignant cells expressing PD-L1 and the densities/frequencies of immune subpopulations analyzed by mIF staining of tumors resected post-ICI therapy, as shown in the table below:

Gut microbiome variable	Immune subpopulations in post-therapy tumors by mIF in patients with nodal non-caseating granulomas	n	Spearman correlation	P value
ISI	Panel 1 Malignant cells PD-L1+ (%)	4	0.633	0.5
ISI	Panel 1 CD3+ T cells (n/mm ²)	4	-0.6	0.417
ISI	Panel 1 CD3+CD8+ T cells (n/mm ²)	4	0.4	0.75
ISI	Panel 1 CD3+PD-1+ T cells (n/mm ²)	4	0.2	0.917
ISI	Panel 1 CD3+CD8+PD-1+ T cells (n/mm ²)	4	0.2	0.917
ISI	Panel 1 Macrophages (n/mm ²)	4	0	1
ISI	Panel 1 Macrophages PD-L1+ (%)	4	0.4	0.75
ISI	Panel 2 CD3+ T cells (n/mm ²)	4	0.4	0.75
ISI	Panel 2 CD3+CD8+ T cells (n/mm ²)	4	0.8	0.333
ISI	Panel 2 CD3+CD8+GZB+ T cells (n/mm ²)	4	0.8	0.333
ISI	Panel 2 CD3+CD8+CD45RO+ T cells (n/mm ²)	4	0.2	0.917
ISI	Panel 2 CD3+CD8-FOXP3+ T cells (n/mm ²)	4	0.2	0.917
Collinsella	Panel 1 Malignant cells PD-L1+ (%)	4	-0.105	1
Collinsella	Panel 1 CD3+ T cells (n/mm ²)	4	-0.4	0.75
Collinsella	Panel 1 CD3+CD8+ T cells (n/mm ²)	4	-0.4	0.75
Collinsella	Panel 1 CD3+PD-1+ T cells (n/mm ²)	4	-0.8	0.333

Collinsella	Panel 1 CD3+CD8+PD-1+ T cells (n/mm ²)	4	-0.8	0.333
Collinsella	Panel 1 Macrophages (n/mm ²)	4	-0.2	0.917
Collinsella	Panel 1 Macrophages PD-L1+ (%)	4	-0.4	0.75
Collinsella	Panel 2 CD3+ T cells (n/mm ²)	4	0.4	0.75
Collinsella	Panel 2 CD3+CD8+ T cells (n/mm ²)	4	0.2	0.917
Collinsella	Panel 2 CD3+CD8+GZB+ T cells (n/mm ²)	4	0.2	0.917
Collinsella	Panel 2 CD3+CD8+CD45RO+ T cells (n/mm ²)	4	0.8	0.333
Collinsella	Panel 2 CD3+CD8-FOXP3+ T cells (n/mm ²)	4	0.8	0.333
Adlercreutzia	Panel 1 Malignant cells PD-L1+ (%)	4	-0.738	0.333
Adlercreutzia	Panel 1 CD3+ T cells (n/mm ²)	4	-0.8	0.333
Adlercreutzia	Panel 1 CD3+CD8+ T cells (n/mm ²)	4	-0.8	0.333
Adlercreutzia	Panel 1 CD3+PD-1+ T cells (n/mm ²)	4	-0.4	0.75
Adlercreutzia	Panel 1 CD3+CD8+PD-1+ T cells (n/mm ²)	4	-0.4	0.75
Adlercreutzia	Panel 1 Macrophages (n/mm ²)	4	-1	0.083
Adlercreutzia	Panel 1 Macrophages PD-L1+ (%)	4	-0.8	0.333
Adlercreutzia	Panel 2 CD3+ T cells (n/mm ²)	4	-0.8	0.333
Adlercreutzia	Panel 2 CD3+CD8+ T cells (n/mm ²)	4	-0.6	0.417
Adlercreutzia	Panel 2 CD3+CD8+GZB+ T cells (n/mm ²)	4	-0.6	0.417
Adlercreutzia	Panel 2 CD3+CD8+CD45RO+ T cells (n/mm ²)	4	-0.4	0.75
Adlercreutzia	Panel 2 CD3+CD8-FOXP3+ T cells (n/mm ²)	4	-0.4	0.75

NIF, nodal immune flare; NIF group includes all available samples with datapoints from patients with pathological evidence of nodal non-caseating granulomas. ISI, Inverse Simpson Index. Two-sided *P* value is from Spearman correlation.

We then investigated the associations between gut microbiome metrics and the gene expression scores of immune cell populations assessed by NanoString in the nodes collected after neoadjuvant ICIs in patients within the NIF group. Eight patients had available resected nodal samples for NanoString gene expression analysis as well as gut microbiome datapoints. Overall, we did not observe significant associations, as assessed by Spearman correlation test, between the gut microbiome metrics and the selected immune cell scores, as shown in the Table below:

Gut microbiome variables	Immune gene expression by NanoString in post-therapy nodes with non-caseating granulomas	n	Spearman correlation	P value
ISI	CD45 cells	8	0.095	0.84
ISI	Macrophages	8	-0.214	0.619
ISI	DCs	8	0.286	0.501
ISI	Cytotoxic cells	8	-0.214	0.619
ISI	Th1	8	-0.143	0.752
ISI	Exhausted CD8 T cells	8	-0.143	0.752
Collinsella	CD45 cells	8	-0.238	0.582
Collinsella	Macrophages	8	-0.262	0.536

Collinsella	DCs	8	-0.048	0.935
Collinsella	Cytotoxic cells	8	-0.071	0.882
Collinsella	Th1	8	-0.786	0.028
Collinsella	Exhausted CD8 T cells	8	0.024	0.977
Adlercreutzia	CD45 cells	8	-0.262	0.536
Adlercreutzia	Macrophages	8	-0.238	0.582
Adlercreutzia	DCs	8	-0.619	0.115
Adlercreutzia	Cytotoxic cells	8	-0.214	0.619
Adlercreutzia	Th1	8	-0.524	0.197
Adlercreutzia	Exhausted CD8 T cells	8	-0.048	0.935

NIF, nodal immune flare; NIF group includes all available samples with datapoints from patients with pathological evidence of nodal non-caseating granulomas. ISI, Inverse Simpson Index. Two-sided *P* value is from Spearman correlation.

Based on the limited number of NIF samples with datapoints available in both analyses, we feel caution should be used in the interpretation of these results which serve as a foundation to build upon for subsequent validation studies in larger NIF clinical cohorts. Adequately powered mechanistic studies will also be needed to examine the relationship between gut microbiome and nodal immune infiltrates and the potential functional role of gut microbiota in the formation of non-caseating granulomas in NIF patients after neoadjuvant ICIs. Therefore, we included the results of these analyses in this response only.

We updated the text of the revised manuscript to state the need for further studies in larger cohorts of NIF patients to determine the a potential causative role of gut microbiome in nodal non-caseating granulomas formation, as detailed below:

Results (pages 20-21):

“These exploratory results suggest that the relative abundance of unique fecal microbiome taxa is associated with NIF, a sarcoid-like inflammatory reaction characterized by de-novo non-caseating granuloma formation after neoadjuvant ICI treatment. Subsequent studies will be needed to validate these associations in larger and annotated NIF clinical cohorts.”

Discussion (page 24):

“The initial associations of relative abundance of unique intestinal microbiota with the NIF phenomenon revealed by our exploratory analyses should serve as a foundation to build upon for subsequent validation in larger clinical NIF datasets and for mechanistic studies aimed to determine the potential functional role of gut microbiota in the formation and composition of nodal non-caseating granulomas in patients developing NIF after neoadjuvant ICIs.”

Minor:

Comment:

1. Fig 1 and Table 2 are difficult to interpret. Classic case of NIF and no NIF should be represented in fig 1 as well as histogram to demonstrate the most important result of the trial 0% vs 13% and 19% NIF in IO neo-adjuvant group.

Response:

Thank you for pointing this out and we apologize if these results were not presented clearly in the prior version of the manuscript. In response to the reviewer's suggestions, we have made the following changes:

In revised Figure 1 we included an illustrative case of a patient with NIF, characterized by abnormal nodes on imaging post-therapy, and absence of cancer and evidence of non-caseating granulomas upon pathological examination from the NEOSTAR cohort (Fig. 1a-f), as well as an illustrative case of a patient with No-NIF, characterized by abnormal nodes on imaging post-therapy, and presence of malignant cells with nodal disease progression and absence of non-caseating granulomas upon pathological examination (Fig. 1g-k).

We have summarized the text reporting the results described in the previous Tables 2 and 3 (merged into a revised Table 2), and in the previous Fig. 2 (now revised Fig. 3) to highlight the key findings of the radiological tumor and nodal changes in patients with abnormal nodes post-therapy from the ICON cohort and the NIF and No-NIF groups from the NEOSTAR cohort.

We have included in the revised Fig. 2 a bar graph illustrating the proportions of patients with NIF characterized by abnormal nodes on imaging post-therapy and the absence of cancer and evidence of non-caseating granulomas upon pathological examination in the overall NEOSTAR cohort (16%), the nivolumab arm (13%) and the nivolumab plus ipilimumab arm (19%), and in the ICON cohort (0%).

These changes are detailed below:

Results (pages 14-15):

“Apparent radiological nodal disease progression after neoadjuvant ICIs in absence of cancer and with non-caseating granulomas

To determine whether the administration of neoadjuvant immunotherapy is associated with unconventional radiological patterns of nodal involvement, we examined the abnormal nodes on scans in patients after they were treated with neoadjuvant ICI therapy on the randomized NEOSTAR study (**Table 1**). We noted that among 44 patients, several patients exhibited abnormal nodes on imaging after treatment that mimicked disease progression but were devoid of cancer upon pathological examination and were instead characterized by the presence of non-caseating granulomas, as shown **Fig. 1a-f**. We referred to this phenomenon as nodal immune flare (NIF). For comparison, an illustrative case of abnormal nodes on imaging after neoadjuvant ICI therapy that was classified as true nodal disease progression based on the presence of malignant cells upon pathological assessment and the lack of non-caseating granulomas is shown in **Fig. 1g-k**. We then questioned whether this pattern of apparent radiological nodal progression in absence of cancer and histological presence of non-caseating granulomas also occurred in patients that were treated with neoadjuvant platinum-based chemotherapy, although empirically we have not been alerted to this phenomenon previously. We analyzed abnormal nodes on imaging after neoadjuvant chemotherapy in a subset of patients from the ICON cohort at our institution (**Table 1**) and did not observe instances where abnormal nodes post-therapy were devoid of cancer and possessed non-caseating granulomas upon pathological examination. These findings suggest that neoadjuvant ICI therapy, compared with platinum-based chemotherapy, is more likely to be associated with unusual radiological appearances of cancer-free lymph nodes that mimic disease progression with the onset of pathological features of sarcoid-like inflammation.”

Figure 1 Legend (pages 38-39):

Figure 1. Radiological and histopathological features of abnormal nodes following neoadjuvant ICIs. a-d, Axial contrast enhanced CT (a), and ¹⁸F-FDG PET/CT (b) images of the mediastinum showing normal nodes prior to neoadjuvant treatment with ICIs on NEOSTAR study in a patient with NSCLC with clinical stage IIIA (metastasis to station 7; stations 4R, 4L, and 11L negative after invasive baseline mediastinal staging with EBUS). ¹⁸F-FDG uptake in the mediastinum is due to esophagitis. Restaging axial CT (c) and ¹⁸F-FDG PET/CT (d) images post neoadjuvant ICIs show marked increase in nodal size and FDG uptake at bilateral mediastinal regions, suspicious for nodal progression. Abnormal nodes were also present at bilateral hilar and right axillary regions (not shown). Mediastinoscopy post-neoadjuvant ICIs did not demonstrate carcinoma in lower paratracheal stations (4L and 4R). **e-f,** Fine needle aspiration (FNA) image of paratracheal nodal station pre-therapy (e) demonstrating lack of tumor cells and normal composition (Papanicolaou, x40), and resected station 4R lymph node post-therapy (f) revealing absence of cancer and evidence of necrotizing non-caseating granulomatous inflammation (hematoxylin and eosin, x10). **g-j,** Axial contrast enhanced CT (g), and ¹⁸F-FDG PET/CT (h) images of the mediastinum show nodal enlargement and abnormal ¹⁸F-FDG uptake in the right hilum and right mediastinum prior to neoadjuvant ICIs on NEOSTAR study in a patient with NSCLC with clinical stage IIIA (baseline invasive mediastinal staging with mediastinoscopy revealed metastasis to station 4R). Restaging axial contrast enhanced CT (i) and ¹⁸F-FDG PET/CT (j) images show increase in size and increase in FDG uptake at right hilar, right mediastinal (4R) and prevascular nodes, consistent with progression of nodal metastasis. Abnormal nodes were also present at mediastinal 1R, 2R and 7 stations, which were previously normal at baseline (not shown). Subsequent biopsy confirmed carcinoma in the right paratracheal (2R and 4R) and subcarinal stations. **k,** FNA image of post-ICI abnormal node (station 7 pictured) revealed the presence of malignancy with disease progression (Papanicolaou, x40). NIF, nodal immune flare; CT, computed tomography; FDG, fluorodeoxyglucose; FNA, fine needle aspiration; PET, positron emission tomography; PD, progressive disease.”

Results (page 16):

“Overall, 16% of patients treated with ICIs on the NEOSTAR study (7/44, 95% CI 7%-30%) were noted to have NIF (**Fig. 2e**). Thirteen percent (3/23, 95% CI 3% – 34%) of cases were observed in the nivolumab monotherapy group, and 19% (4/21, 95% CI 5% – 42%) were seen in those treated with nivolumab plus ipilimumab (**Fig. 2e**). No cases of NIF were observed in patients treated with neoadjuvant platinum-based chemotherapy in the ICON cohort (0%, 0/28) (**Fig. 2e**).”

Figure 2 Legend (page 39):

“**Figure 2. Histopathological features of nodal specimens pre- and post-neoadjuvant ICIs and chemotherapy in NEOSTAR and ICON patients. a,** Illustrative fine needle aspiration (FNA) image from preoperative mediastinal staging by EBUS in NEOSTAR NIF patient did not demonstrate granulomatous inflammation within examined nodes (station 4L pictured; Papanicolaou, x40). **b,** Resected nodal specimen in NEOSTAR NIF patient following ICIs demonstrating a diffuse non-caseating granulomatous inflammatory reaction (station 11R pictured; hematoxylin and eosin, x10). **c,** Illustrative FNA image from preoperative mediastinal staging by EBUS in ICON No-NIF patient did not demonstrate granulomatous inflammation within examined nodes (station 7 pictured; Papanicolaou, x40). **d,** Resected nodal specimen following neoadjuvant chemotherapy in a patient with no-NIF from ICON cohort with absence of diffuse non-caseating granulomatous inflammatory reaction (station 7 pictured; hematoxylin and eosin, x4). **e,** Proportions of patients with NIF, characterized by abnormal nodes on imaging that are devoid of cancer and contain *de novo* non-caseating granulomas in NEOSTAR (n=44) and ICON (n=28) patient cohorts. EBUS, endobronchial ultrasound; FNA, fine needle aspiration.”

Results (pages 17-18):

“Changes in radiological tumor and nodal parameters after neoadjuvant chemotherapy and ICIs

To determine the changes in the radiologic features of tumor and lymph nodes induced by neoadjuvant chemotherapy and ICIs, we measured the tumor volume and SUV_{max} and the nodal size and SUV_{max} in both ICON and NEOSTAR patients with abnormal nodes post-therapy and compared these values to their baseline measurements. In ICON patients, the median tumor volume decreased from 68.2 to 20.6 mL ($P = 0.016$) and the median tumor SUV_{max} decreased from 9.30 to 8.90 after chemotherapy, although this did not reach statistical significance (**Table 2**). In the same cohort, the mean nodal size decreased from 1.54 cm at baseline (pre-therapy) to 1.25 cm after chemotherapy ($P = 0.058$) (**Fig. 3a**), whereas the mean nodal SUV_{max} remained unchanged (**Fig. 3b**). In NEOSTAR patients with NIF, the mean tumor volume and SUV_{max} did not significantly change after ICIs (**Table 2**). However, the mean size and SUV_{max} of abnormal nodes increased from 0.91 cm to 1.20 cm ($P < 0.001$; **Fig. 3c**) and from 2.82 to 6.33 ($P < 0.001$; **Fig. 3d**), respectively. In NEOSTAR patients in the No-NIF group, the mean tumor volume and SUV_{max} did not significantly change after ICI therapy (**Table 2**), whereas both the mean size ($P = 0.013$; **Fig. 3e**) and SUV_{max} ($P < 0.001$; **Fig. 3f**) of abnormal nodes increased after ICI therapy. While nodal size and SUV_{max} increased in both NIF and No-NIF groups in the NEOSTAR cohort, the magnitude of change of both parameters was greater in patients with NIF as compared to that of patients with No-NIF (mean nodal size difference, $P = 0.139$; **Fig. 3g**; mean nodal SUV_{max} difference, $P < 0.001$; **Fig. 3h**). Taken together, these results suggest that nodal size and SUV_{max} can increase after neoadjuvant immunotherapy in both NIF and No-NIF patients, with a greater increase in nodal SUV_{max} in NIF patients, without significant changes in tumor volume and tumor SUV_{max} and illustrate the importance of pathological examination of abnormal nodes on imaging after ICIs.”

Table 2 (page 37):

“Table 2. Changes in tumor volume and SUV_{max} in ICON and NEOSTAR patients with abnormal nodes post-therapy.

ICON cohort	Pre-chemotherapy			Post-chemotherapy			P value
Tumor volume (mL)* Median (min, max)	n=8 68.2 (2.50, 207)			n=8 20.6 (2.60, 94.0)			
Change (Post-Pre) Median (min, max)	n=8 -35.9 (-113, 0.10)						0.016
Tumor SUV_{max}** Median (min, max)	n=3 9.30 (7.10, 19.0)			n=3 8.90 (6.50, 9.60)			
Change (Post-Pre) Median (min, max)	n=3 0.30 (-14.4, 1.80)						1.00
NEOSTAR cohort	NIF		P value	No-NIF		P value	P value
	Pre-ICIs	Post-ICIs		Pre-ICIs	Post-ICIs		
Tumor volume (mL) Mean (SD)	n=7 23.8 (8.60)	n=7 17.1 (13.7)	0.496	n=18 49.0 (50.9)	n=18 37.1 (44.5)	0.059	
Change (Post-Pre) Mean (SE)	n=7 -6.67 (9.65)			n=18 -12.0 (6.02)			0.089

Tumor SUV_{max}[^] Mean (SD)	n=7 10.4 (4.40)	n=7 8.98 (6.56)	0.643	n=17 14.4 (6.93)	n=17 14.7 (9.05)	0.867	
Change (Post-Pre) Mean (SE)	n=7 -1.39 (2.95)			n=17 0.32 (1.89)			0.142

ICON, ImmunogenomiC prOfiling in Non-small Cell Lung Cancer; *Post-chemotherapy tumor volume was not measurable (atelectasis) in one patient. **Six patients did not have paired tumor SUV_{max} values. ^One patient did not have post-ICI PET/CT images. SUV, standardized uptake value. SD, standard deviation; SE, standard error. In the ICON cohort, the two-sided *P* value is from Wilcoxon signed-rank test for paired tumor volume and tumor SUV_{max} comparisons. In the NEOSTAR cohort, the two-sided *P* value is from linear mixed-effect model for tumor size and tumor SUV_{max} comparisons.”

Figure 3 Legend (pages 39-40):

“**Figure 3. Changes in nodal size and SUV_{max} in ICON and NEOSTAR patients with abnormal nodes post-therapy.** **a**, Mean nodal size (cm) of abnormal nodes post neoadjuvant chemotherapy as compared to pre-therapy in ICON patients. Data is shown as mean nodal size in cm ± SD. Two-sided *P* value is from linear mixed-effect model. N₁ = number of nodes analyzed in nine patients. N₂ = number of nodes analyzed in nine patients. **b**, Mean nodal SUV_{max} of abnormal nodes post neoadjuvant chemotherapy as compared to pre-therapy in ICON patients. Data is shown as mean nodal SUV_{max} ± SD. Two-sided *P* value is from linear mixed-effect model. N₁ and N₂ = number of nodes analyzed in three patients. **c, d**, Mean nodal size (c) and SUV_{max} (d) of abnormal nodes post neoadjuvant ICIs as compared to pre-therapy in NEOSTAR patients with NIF. Data is shown as mean nodal size in cm ± SD. N₁ and N₂ = number of nodes analyzed in seven patients. **e, f**, Mean nodal size (e) and SUV_{max} (f) of abnormal nodes post neoadjuvant ICIs as compared to pre-therapy in NEOSTAR patients with No-NIF. Data is shown as mean nodal size in cm ± SD. N₁ and N₂ = number of nodes analyzed in 17 patients (e) and 15 patients (f) with available scans/images. **g**, Difference in mean size of abnormal nodes between post- and pre-therapy in NEOSTAR patients with NIF as compared with those with No-NIF. Data is shown as change in mean nodal size in cm ± SE. N₁ = number of nodes analyzed in seven patients. N₂ = number of nodes analyzed in 17 patients. Two-sided *P* value is from linear mixed-effect model. **h**, Difference in mean SUV_{max} of abnormal nodes between post- and pre-therapy in NEOSTAR patients with NIF as compared with those with No-NIF. Data is shown as change in mean nodal SUV_{max} ± SE. N₁ = number of nodes analyzed in seven patients. N₂ = number of nodes analyzed in 15 patients. Two-sided *P* value is from linear mixed-effect model. SUV, standardized uptake value; SD, Standard Deviation; SE, Standard Error.”

Comment:

2. The paragraph on probabilities of true vs false NIF is extremely confusing with no table or explanatory figure.

Response:

We added a figure of probabilities of observing 0 incidence of NIF (Supplementary Fig. 1) to facilitate visualization of the data reported in the paragraph describing the probabilities of no cases of NIF in the ICON chemotherapy cohort in the Result section, as shown below:

Results (page 16):

“To evaluate the likelihood, by chance, of identifying no cases of NIF in chemotherapy-treated ICON patients, we calculated the probabilities of observing no NIF cases assuming several true incidence rates. The probabilities of observing 0 incidence of NIF in the 28 ICON patients were 23.8%, 5.2%,

and only 0.8% if the true NIF rates were 5%, 10%, and 16%, respectively (**Supplementary Fig. 1**). We further calculated the posterior probability that the NIF rate in the ICON cohort (p) was equal to or greater than the observed NIF rate in the NEOSTAR cohort (16%) and a prior p beta (0.16, 0.84), indicating a prior belief that the NIF rate in the ICON cohort was the same as the one in the NEOSTAR cohort. This probability was only 0.04%.”

Supplementary Figure 1 Legend (page 8):

“**Supplementary Figure 1. Probability of NIF incidence in ICON cohort.** The probabilities of observing 0 incidence of NIF in the 28 ICON patients were 23.8%, 5.2%, and only 0.8% if the true NIF rates were 5%, 10%, and 16%, respectively. The calculations were performed using the binomial distribution.”

Comment:

3. Figure 2 would benefit from a different color codes or shape for each category to make it easier to read.

Response:

Thank you for the suggestion. We modified all graphs displayed in the revised Figure 3 (previously Figure 2) using different shapes (circles and squares) for each timepoint (pre-therapy and post-therapy) and colors (red and blue) for each category (NIF and No-NIF) to improve the visualization of the results.

Reviewer #3 (Remarks to the Author): with expertise in biostatistics - clinical trials

This is an exploratory analysis examining the association of nodal immune flare and response. The methods used are both appropriate and well described. I do however have some comments on the methods used.

Comment

a) The authors claim to use Fisher's exact test to test categorical variables although (i)I could not identify where this was used and (ii) many biostatisticians (strongly) recommend against its use (except perhaps in very specialised cases) - D'Agostino et al, Amer Stat 1987, Lydersen et al Stat in Med, 2009; Berkson, J Stat Plan & Infer 1978 to quote just a few.

Response:

We appreciate the reviewer's comments. We understand that Fisher's exact test yields conservative P values. Per reviewer's recommendation, we changed the Fisher's exact test to the unconditional exact test using a Z-pooled statistics suggested by Suissa and Shuster (R package: Exact). The test was applied for the analyses performed in **Supplementary Tables 1, 3, 5 and 6**. We mentioned the type of test used for the P value indicated in the caption of the Supplementary tables. We have also updated the methods section to reflect this change, as follows:

Methods (page 14):

“Unconditional exact test available in the R package “Exact” was used to compare the categorical distributions between patient populations.”

Comment:

b) The Wilcoxon test DOES NOT test medians - I'm surprised that the statistical authors allowed this approach. Again there are numerous examples where medians are equal and the Wilcoxon test is significant [because it tests a shift in an (undefined) location of the two distributions.

A simple example

Group A: 0 2 7 Group B: 1 2 15 Median Group A/B 3

Freq 20 20 20 20 20 Rank-sum p-val 0.036

Response:

The Wilcoxon rank sum test does not test for equal median in the general setting. Testing for equal median is valid only in the location shift model. In response to the reviewer's comment, we modified the statement describing the use of this statistical method in the Methods section of the revised manuscript, as follows:

Methods (page 14):

"The Wilcoxon rank sum test was used to compare the distributions of continuous variables and immune markers between patient populations."

Reviewer #4 (Remarks to the Author): with expertise in cancer immunotherapy and imaging

The manuscript aims to assess the role of NIF evaluated by using molecular imaging such as PET/CT with FDG in a small cohort of patients undergoing ICIs neoadjuvant therapy for NSCLC.

The manuscript is interesting, being the evaluation of response to therapy with ICIs a very hot topic, but from the imaging point of view it appears very poor.

Please below some comments:

Comment:

1) The rate of pseudo progression in NSCLC from ICIs therapy should be described in the Introduction section rather than in the discussion part.

Response:

We have revised the introduction section to incorporate the description of the rate of pseudo-progression in NSCLC patients treated with immune checkpoint inhibitors, as detailed below:

Introduction (page 5):

"One potential concern with ICIs is tumor pseudo-progression, the appearance of tumor growth without true progressive disease (PD) thought to be due to increased intratumoral immune cell infiltration, which has been reported in patients with NSCLC at a rate ranging between 0.6% and 5.8%^{7,8,9}."

Discussion (page 22):

"The novel mechanisms of action of ICIs have prompted reconsideration of the patterns of radiological responses to therapy³⁶. The NIF phenomenon reported here is distinct from tumor pseudo-progression in which a tumor initially progresses and later responds to therapy^{9,37}. Here, the lymph nodes alone appeared to progress radiographically, yet the tumor remained stable or became smaller. The nodal enlargement and increased metabolic activity characterizing apparent cancer progression are due to nodal inflammation and pathological features of non-caseating granulomas."

Comment:

2) the assumption of SUV_{max} equal to 4.5 in the Introduction paragraph should be justified or enriched with a reference.

Response:

The SUV_{max} cut-off value of ≥ 4.5 was used to characterize abnormal nodes suspicious for malignancy on images. The description of the selected radiological parameters to define abnormal nodes radiologically has been revised in the Methods session, "Radiological assessment", to include justification and enriched with supporting references as suggested, and as detailed below:

Methods (page 8):

"As described above, on ¹⁸F-FDG PET/CT a SUV_{max} cutoff value of ≥ 4.5 was used to characterize abnormal nodes suspicious for malignancy. Although the traditional SUV_{max} cut-off of 2.5 is associated with a lower false negative rate, which is important to avoid missing nodal metastasis, higher SUV_{max} cut-offs have been shown to improve PET/CT diagnostic performance in NSCLC nodal staging by decreasing the false positive rate and may be helpful in distinguishing malignant from inflammatory nodes^{15, 16, 17, 18, 19.}"

Comment:

3) The study by Cascone et al in the methods section should be reported as reference.

Response:

We have incorporated the study reporting the primary, and select secondary exploratory outcomes of the NEOSTAR randomized study by Cascone et al. in *Nat Med* 2021 (PMID: 33603241) in the Methods session and throughout the revised manuscript, where appropriate, as ref 12.

Comment:

4) the final number of enrolled patients is unclear. The sentences of page 7 lines 122-127 are very confused. I am not able to understand if the enrolled patients are 37 or 35 or 36. Please specify.

Response:

Thank you for allowing us to clarify this point. In the randomized NEOSTAR study, overall 39 of 44 treated patients (ITT population) underwent surgical resection; among these, 37 patients were resected on trial after ICI therapy, whereas two patients who received at least one dose of ICI therapy on trial underwent surgery off trial after receiving additional systemic therapies (one due to concern of radiological PD post ICIs which was instead a NIF case, and one due to the development of TRAE prior to radiological restaging on trial). For the radiological analysis of the NEOSTAR cohort, all available images in 44 patients were analyzed. Measurements of abnormal nodal size and SUV_{max} were recorded in 24 and 22 patients post-neoadjuvant ICI therapy, respectively and compared to corresponding pre-therapy nodal size and metabolic activity.

The final pathological node analysis to assess the presence of cancer or non-caseating granulomas after neoadjuvant therapy was performed in all available post-therapy samples from the NEOSTAR cohort (n=41: 37 resected on trial, two resected off trial and two not resected that had available fine needle aspirate [FNA] material of abnormal nodes post-therapy). Examination of pre-therapy lymph nodes consisted of review of available FNA/histological material obtained during invasive mediastinal staging procedures.

For the analyses evaluating the association of NIF with radiographic responses, MPR and TRAEs, status, we included the ITT of 44 NEOSTAR patients. For the analyses evaluating the association of

NIF with the percentage viable tumor at resection, we included 37 NEOSTAR patients resected on trial after the administration of ICIs.

Given the overall modest number of NIF cases with post-therapy tumor and nodal immune infiltrate datapoints for the immune correlative studies performed with immunohistochemistry, multiplex immunofluorescence and NanoString, we included in the NIF group of these analyses the datapoints from all available post-therapy tissue samples of patients with nodal non-caseating granulomas.

The immunohistochemical staining and analysis were performed in resected tumor tissues from six NIF and 22 No-NIF patients. The mIF staining and analysis were performed in resected tumor tissues from four NIF and 21 No-NIF patients. The nodal transcriptomic analyses were performed in non-cancerous post-therapy nodes resected from eight NIF and 29 No-NIF patients.

The microbiome diversity and composition analyses were performed in seven NIF and 29 No-NIF patients.

We have revised the **Methods section (pages 9-13)** of the resubmitted manuscript to clarify the numbers of patients enrolled on the study and the number of samples analyzed for each analysis reported in this manuscript.

Comment:

5) the final population for the chemotherapy arm is 22 and not 28, as reported in page 7, differently from the result paragraph.

Response:

We apologize for the lack of clarity on the number of ICON patients included in our analyses. For the radiological analysis of the ICON cohort, all available images in 28 patients were analyzed. Measurements of abnormal nodal size and SUV_{max} were recorded in nine and three patients post-neoadjuvant chemotherapy, respectively, and compared to corresponding pre-therapy nodal size and metabolic activity.

The final pathological node analysis to assess the presence of cancer or non-caseating granulomas after neoadjuvant therapy was performed in all tissue specimens available for analysis in resected patients from the ICON cohort (n=22).

We have revised the **Methods section (pages 7 and 9)** of the resubmitted manuscript to clarify the numbers of ICON patients analyzed.

Comment:

6) data about CT and PET interpretation and also acquisition protocol and timing from the ICIs administration is completely missing. Please add some additional sentences. The time and the number of cycles can significantly affect the interpretation of the molecular imaging.

Response:

As suggested, we have revised the Methods section to incorporate a detailed description of CT and PET/CT imaging acquisition protocol, imaging findings interpretation, and timing of imaging acquisition from the administration of ICIs on study in the Methods session of the revised manuscript, as shown below:

Methods (pages 7-8; imaging acquisition details):

“CT scans were acquired in a multidetector scanner following IV contrast administration unless contraindicated. Multiplanar CT image series were reconstructed with 2.5 mm slice thickness using standard and high spatial reconstruction algorithms. FDG-PET/CT imaging was performed using Discovery STE PET/CT scanner (GE healthcare Waukesha WI). All patients fasted for 6 hours before the FDG injection and had confirmed normal fasting blood glucose level of less than 200 mg/dL. PET was performed in 3-dimensional mode at 3 – 5 min per bed station depending on patient BMI. An intravenous injection of 9-11 mCi of FDG was administered in the arm or central venous catheter on the side opposite to the cancer, and emission scans were acquired at 70 ± 10 minutes after the FDG injection. The acquired PET data were corrected for scatter coincidences, random coincidences, deadtime, and attenuation and reconstructed using OSEM on standard vendor-provided workstations. Non-contrast-enhanced CT images from the base of the skull to the mid-thigh were acquired in helical mode (speed, 13.5 mm per rotation) during shallow breathing at a 3.75-mm slice thickness, a tube voltage of 120 kVp, and 0.5-second rotation with tube current modulation. Daily quality control procedures were performed on all PET scanners to ensure cross-calibration between systems and normalize differences in system performance. In a small number of patients, CT scan or PET/CT were performed at outside institution with comparable technique.”

Methods (page 8; imaging findings interpretation):

“All available CT and PET/CT images were reviewed in all patients in both cohorts by a board-certified thoracic radiologist. Measurements of short-axis diameter on CT and ^{18}F -FDG maximum standardized uptake value (SUV_{max}) were recorded for all abnormal mediastinal or hilar nodes, which were defined as nodal short axis diameter > 1.0 cm (> 1.2 cm in the subcarinal region)¹⁴ on CT images and/or SUV_{max} ≥ 4.5 on PET/CT images post-therapy and compared to their corresponding pre-therapy measurements. Measurements were also obtained in any abnormal extra thoracic node if it met CT and ^{18}F FDG uptake inclusion criteria described above. Tumor volume measurement was performed using a commercially available semi-automatic software MIM version 6.6.6 (MIM Software Inc., Cleveland, Ohio). Characterization of nodal size on CT was performed by measuring short axis diameter using mediastinal window setting (level = 50; width = 350). Characterization of tumor and mediastinal lymph node ^{18}F -FDG uptake was performed using semiquantitative analysis of the SUV_{max} (MIM version 6.6.6; MIM Software Inc., Cleveland, Ohio). In a few cases, inclusion of outside PET/CT scans or the lack of available images precluded SUV_{max} measurements. As described above, on ^{18}F -FDG PET/CT a SUV_{max} cutoff value of ≥ 4.5 was used to characterize abnormal nodes suspicious for malignancy. Although the traditional SUV_{max} cut-off of 2.5 is associated with a lower false negative rate, which is important to avoid missing nodal metastasis, higher SUV_{max} cut-offs have been shown to improve PET/CT diagnostic performance in NSCLC nodal staging by decreasing the false positive rate and may be helpful in distinguishing malignant from inflammatory nodes^{15, 16, 17, 18, 19}. Positive findings were compared with the post-therapy cytopathologic/histopathological specimen findings and pathological staging for all patients.”

Methods (page 6; timing of imaging acquisition):

“NEOSTAR (NCT03158129) is a single-institution, investigator-initiated phase II trial on which the first 44 patients with stage I-III A (N2 single station; 7th edition of the American Joint Committee on Cancer [AJCC]) operable NSCLC were randomized after baseline CT and ^{18}F -FDG PET/CT to receive three doses of neoadjuvant PD-1 inhibitor nivolumab (3 mg/kg intravenously [IV] on days 1, 15, and 29) or combination of nivolumab (3 mg/kg IV every 2 weeks, on day 1, 15 and 29) plus the CTLA-4 inhibitor ipilimumab (1 mg/kg IV every 6 weeks, on day 1 only) followed by restaging CT and PET/CT imaging (both recommended at least 14 days after last dose of ICI therapy) and subsequent surgical resection¹².”

Comment:

7) for the interpretation of metabolic lymph nodes, the authors used a very old paper (2006) that considered only a PET scanner and not an hybrid scanner. Therefore, the authors should use more appropriate and recent criteria, please check the following papers (PMID: 22459646 or 32525121).

Response:

Thank you for pointing this out. Over the last 20 years numerous activity thresholds for lymph node positivity have been explored in the literature, including use of SUV_{max} 2.5 or higher as the cut-off, and indexes comparing nodal activity with liver background, mediastinal blood pool (MBP), and tumor SUV_{max}. As mentioned by the reviewer, the comparative indices have been proposed more recently; however, despite their availability for several years, they have not been incorporated into clinical routine of PET/CT interpretation, possibly for being less practical for daily use. There is data supporting that all the above-mentioned parameters show statistically significant differences between malignant and benign lymph nodes and can be used in lung cancer staging (PMID: 24900137). The combination of the SUV_{max} and size of lymph node has been preferred in most institutions.

In this manuscript, the authors opted to use the SUV_{max} for characterization of lesions that were visually abnormal (showing FDG uptake above the MBP), which is how PET/CT scans are interpreted in our clinical practice. Although the traditional SUV_{max} cut-off of 2.5 is still frequently used because it provides a low FNR, higher PET/CT cut-offs have been shown to improve diagnostic performance in NSCLC nodal staging. This is particularly important when trying to decrease the number of false positive results, such as in the context of possible inflammatory nodes. An optimal SUV_{max} cut-off has not been well defined, with studies demonstrating the ideal cut-off to vary from 2.5 to 5.3 (References: PMID: 17942814, PMID: 16863739, PMID: 24900137, PMID: 28658052, PMID: 31420186). In our study, we selected a SUV_{max} cut-off of 4.5 based on a study by Hellwig et al, which evaluated 371 biopsy proven lymph nodes and demonstrated a decrease in false positive rate with the highest accuracy at this cut-off level (PMID: 17942814).

As detailed in the session cited from the revised manuscript in response to prior comment, we have incorporated the updated references and the details on imaging interpretation, as detailed above, in the **Methods session (page 8)** of the revised manuscript).

Comment:

8) results section is complete, although the first paragraph is very hard to read, mainly for the statistical assumptions.

Response:

In response to this comment, and to a similar observation made by Reviewer 2, we have summarized the text of the **first paragraph of the Results section** reporting the results described in the previous Tables 2 and 3 (**merged into a revised Table 2**), and in the previous Fig. 2 (now **revised Fig. 3**) to highlight the key findings of the radiological tumor and nodal changes in patients with abnormal nodes post-therapy from the ICON cohort and the NIF and No-NIF groups from the NEOSTAR cohort. Several statistical details have been removed from the revised text describing the results and instead placed in the **revised Table 2 and Fig. 3**.

Comment:

9) the authors should underline that the imaging can miss the presence of NIF in 7% of cases, but it cannot alter the final effect of the ICIs treatment.

Response:

We emphasized in the revised manuscript that 3 additional patients were found to have nodal non-caseating granulomas on pathological examination post-ICI therapy in absence of radiologically abnormal nodes on post-therapy imaging. It is noteworthy that we did not detect a significant association between NIF and radiographic and pathological responses, suggesting that cellular and molecular mechanisms of systemic immunity driving NIF and the antitumor immune responses may be distinct from one another. This point has been described and discussed in the Result and Discussion sections of the revised manuscript, as detailed below:

Results (pages 16-17):

“Interestingly, we also noted that an additional 7% (3/44) of patients who were treated with ICIs on trial had de novo nodal non-caseating granulomas on pathological analysis after treatment but did not exhibit radiologically abnormal nodes post-therapy. The observation of de novo non-caseating granulomas related to all examined lymph nodes, albeit in the absence of nodes suspicious for cancer on imaging post-therapy in some cases, argues for a systemic effect of ICIs on lymph nodes. These findings suggest that the incidence of de novo non-caseating granulomatous nodal inflammation could be as high as 23% (10/44) following at least one cycle and up to three cycles of neoadjuvant ICIs; however, not all cases were associated with false positive radiological evidence of nodal disease progression. Considering these observations, we suggest pathological analysis of any lymph nodes deemed suspicious for disease by standard radiological criteria due to the possibility of false positive findings on restaging imaging.”

Discussion (page 21):

“The incidence of NIF on restaging scans following neoadjuvant ICIs in our study was 16% (7/44). However, we noted the presence of pathologic de novo nodal non-caseating granulomatous inflammation occurred in up to 23% of patients in our cohort (10/44). This rather high observed rate of this phenomenon calls for an extra vigilance on restaging of patients undergoing ICI therapy.”

Discussion (page 23):

“It is noteworthy that we did not detect a significant association between NIF and radiographic and pathological responses, suggesting that cellular and molecular mechanisms of systemic immunity driving NIF and the antitumor immune responses may be distinct from one another.”

Comment:

10) please discuss more in details the information provide in the page 18 lines 394-397. Is the NIF correlated with an under expression of PDL-1? this concept is not clear.

Tables and figures are OK.

Response:

Thank you for giving us the opportunity to detail and clarify the information regarding the correlation between the NIF phenomenon and the expression of PD-L1.

As detailed in response to Reviewer 2, in the current study we found no differences in the percentages of tumor cells expressing PD-L1 as assessed by IHC and mIF staining and in the percentages of macrophages expressing PD-L1 by mIF staining in tumor tissues resected after neoadjuvant ICI therapy in tumor tissues containing non-caseating granulomas as compared with tumor tissues from No-NIF patients, although these results may have been influenced by the modest number of samples available for the analyses.

In the revised Discussion, we clarified that in previous reports, pathological assessment of enlarged and avid nodes revealed sarcoid-like changes and elevated expression of PD-L1 on peripheral immune cells upon discontinuation of ICI therapy in NSCLC³⁸. The investigators found that an elevated expression of PD-L1 by peripheral blood mononuclear cells was associated with nodal and skin sarcoid-like reaction after nivolumab was discontinued in a case of unresectable NSCLC³⁸. The authors speculated that the increase in peripheral PD-L1 may result from cytokines produced by activated immune cells present in sarcoid lesions and/or in the periphery^{38,39}. In the current study, we did not observe greater frequencies of PD-L1+ tumor cells or macrophages within resected tumors of patients with nodal non-caseating granulomas after neoadjuvant ICIs. However, these results may have been influenced by the modest number of samples available for analysis and should be validated in larger studies.

This point has been clarified in the **Discussion section (pages 22-23)** of the revised manuscript.

We thank you again for your valuable comments regarding our work. It is our hope the substantial revisions and clarifications have appropriately addressed your concerns.

Sincerely,

Tina Cascone, M.D., Ph.D. and John V. Heymach, M.D., PhD.

Reviewers' Comments:

Reviewer #1:

Remarks to the Author:

The authors have adequately addressed the majority of the reviewers' comments, thus the paper may be now suitable for publication.

Reviewer #2:

Remarks to the Author:

Thank you for carefully addressing the issues on the immune (new fig 4) and microbiome differences between NIF and no-NIF patients. We clearly see a difference in beta-diversity in both groups and the overrepresentation of Collinsella opening future mechanistic studies.

Reviewer #3:

Remarks to the Author:

The authors have addressed concerns.

Val GebSKI

Reviewer #4:

Remarks to the Author:

The manuscript is significantly improved and therefore it would be accepted.